# IAPP toxicity activates HIF1α/PFKFB3 signaling delaying β-cell loss at the expense of β-cell function

Chiara Montemurro [1,5], Hiroshi Nomoto [1,5], Lina Pei[1,5], Vishal S. Parekh[2], Kenny E. Vongbunyong[1], Suryakiran Vadrevu[2], Tatyana Gurlo[1], Alexandra E. Butler[1], Rohan Subramaniam[1], Eleni Ritou[3], Orian S. Shirihai[3], Leslie S. Satin[2], Peter C. Butler[1] & Slavica Tudzarova [1,4]

The islet in type 2 diabetes (T2D) is characterized by amyloid deposits derived from islet amyloid polypeptide (IAPP), a protein co-expressed with insulin by β-cells. In common with amyloidogenic proteins implicated in neurodegeneration, human IAPP (hIAPP) forms membrane permeant toxic oligomers implicated in misfolded protein stress. Here, we establish that hIAPP misfolded protein stress activates HIF1α/PFKFB3 signaling, this increases glycolysis disengaged from oxidative phosphorylation with mitochondrial fragmentation and perinuclear clustering, considered a protective posture against increased cytosolic $Ca^{2+}$ characteristic of toxic oligomer stress. In contrast to tissues with the capacity to regenerate, β-cells in adult humans are minimally replicative, and therefore fail to execute the second pro-regenerative phase of the HIF1α/PFKFB3 injury pathway. Instead, β-cells in T2D remain trapped in the pro-survival first phase of the HIF1α injury repair response with metabolism and the mitochondrial network adapted to slow the rate of cell attrition at the expense of β-cell function.

---

[1] Larry L. Hillblom Islet Research Center, David Geffen School of Medicine, University of California, Los Angeles, Los Angeles, CA 90024, USA. [2] Department of Pharmacology and Brehm Diabetes Research Center, University of Michigan, Ann Arbor, MI 48105, USA. [3] Division of Endocrinology, Department of Medicine, David Geffen School of Medicine, University of California, Los Angeles, Los Angeles, CA 90024, USA. [4] Jonsson Comprehensive Cancer Center, David Geffen School of Medicine, University of California, Los Angeles, Los Angeles, CA 90024, USA. [5] These authors contributed equally: Chiara Montemurro, Hiroshi Nomoto, Lina Pei. Correspondence and requests for materials should be addressed to S.T. (email: STudzarova@mednet.ucla.edu)

Type 2 diabetes (T2D) is characterized by a progressive defect in insulin secretion in the setting of relative insulin resistance[1]. Histopathological studies in individuals with T2D reveal a partial β-cell deficit with islet amyloid derived from islet amyloid polypeptide (IAPP), a protein co-expressed and co-secreted with insulin by β-cells[2,3]. In common with other protein misfolding diseases, the most toxic forms of IAPP aggregates are small membrane permeant oligomers[4–6]. These oligomers are considered the likely cause of aberrant $Ca^{2+}$ signaling manifest as calpain hyperactivation characteristic of cells affected by misfolded proteins[7–10].

One of the unexplained observations in both T2D and neurodegenerative diseases is the remarkably slow rate of β-cell and neuron loss given the known cytotoxicity of aberrant $Ca^{2+}$ signaling mediated through mitochondrial-induced apoptosis. Another unexplained observation is the shared alterations in metabolism and mitochondrial network changes in affected β-cells and neurons[11–14]. These include increased flux through glycolysis with the diversion of pyruvate to lactate, fragmentation of the mitochondrial network with decreased glucose-induced oxidative phosphorylation, changes that contribute to β-cell, and neuron dysfunction. We noted that the latter changes are also characteristics of cancer cells[15,16], well adapted for survival, implying a potential shared pro-survival signaling pathway. Given this, we paid particular attention to signaling pathways that regulate metabolism in cancer to test the hypothesis that comparable signaling is activated in stressed β-cells.

In the present study, we establish that the conserved HIF1α/PFKFB3 signaling pathway is activated by IAPP misfolded protein-driven stress in pancreatic β-cells to trigger an adaptive protective metabolic response that slows β-cell death at the expense of β-cell function. Further, we establish that this signaling pathway is activated in β-cells in humans with T2D providing a basis for slow β-cell loss.

## Results

**HIF1α/PFKFB3 activation by human IAPP toxicity**. There are numerous hypotheses as to what initiates β-cell dysfunction in T2D. Since the metabolic alterations in β-cells in T2D are similar to those in neurons impacted by misfolded protein stress[17,18], we focused on human IAPP (hIAPP) misfolding induced stress. To identify potential metabolic signaling pathways induced by hIAPP toxicity, we first analyzed available microarray data[19] (GEO Accession Number GSE90779) from islets isolated from human IAPP (HIP) transgenic rats at 4.5 months of age when β-cell stress is present but preceding diabetes onset.

The pattern of changes in metabolic regulatory genes induced by hIAPP toxicity was indeed reminiscent of that in cancer cells, with upregulation of genes involved in glycolysis (lactate dehydrogenase C, LDHC; phosphofructokinase L, PFKL; pyruvate kinase M2, PKM2 and 6-phosphofructo-2-kinase fructose 2,6 biphosphatase, PFKFB3) and downregulation of those engaged in the TCA cycle (pyruvate carboxylase, PC; malate dehydrogenase, MDH; fumarate hydratase, FH; succinate dehydrogenase, SDH) (Fig. 1).

HIF1α, a key regulator of metabolism in cancer[20] as well as in response to stress[21], was upregulated while its inhibitor, Von-Hippel Lindau tumor suppressor, was downregulated (Fig. 1a). Consistent with this, HIF1α target genes were differentially expressed between islets with hIAPP-induced stress and controls (Fig. 1a). PFKFB3, a master regulator of glycolysis[22], was highly upregulated by hIAPP stress (20- and 9.5-fold in two independent experiments). In further support of hIAPP-induced increased aerobic glycolysis, transcription of both LDHA and MCT1, as evaluated by qRT-PCR, was increased in islets from prediabetic

HIP rats ($p < 0.05$, Fig. 1b) as was the rate of lactate production ($p < 0.05$, Fig. 1c). To further test whether hIAPP toxicity induces the HIF1α/PFKFB3 stress/repair signaling pathway, we confirmed that HIF1α and PFKFB3 protein levels were increased in HIP rat islets (Fig. 1d, e). To assure that this pathway was activated well before any confounding effects of hyperglycemia, we evaluated islets from HIP rats at 2.5 and 3.5 months of age, and found that PFKFB3 levels were already increased by 2.5 months of age, namely before frank hyperglycemia (Supplementary Fig. 1a–c and Supplementary Table 1).

**HIF1α/PFKFB3 activation in β-cells of humans with T2D**. Having established the HIF1α/PFKFB3 stress/repair signaling pathway as a potential mediator of hIAPP-induced changes in β-cell metabolism, we next sought to establish if the key mediator of this response, PFKFB3 is increased in β-cells in humans with T2D. We immunostained human pancreas sections from brain dead organ donors procured from the nPOD consortium[23] with T2D and age and BMI matched non-diabetics (ND) and, for comparison, pancreas of HIP and WT rats. The frequency of β-cells immune-positive for PFKFB3 was increased ($p < 0.05$ and $p < 0.001$, respectively) in both humans with T2D and HIP rats (Fig. 2 and Supplementary Table 2). Concomitant with this finding, nuclear HIF1α and PFKFB3 levels were increased in isolated islets from humans with T2D (Fig. 2d and Supplementary Table 3).

To establish if the induction of PFKFB3 in β-cells is under the transcriptional control of HIF1α, we transfected a β-cell line (INS 832/13 cells) with a PFKFB3 promoter-luciferase reporter construct, containing two hypoxia response elements, and tested its activation in control and cells transduced with hIAPP adenoviral vector. Luciferase activity was increased ($p < 0.05$) in INS 832/13 cells overexpressing hIAPP (Fig. 3a), affirming binding of HIF1α to the PFKFB3 promoter.

Furthermore, HIF1α silencing in both INS 832/13 cells and human islets transduced with hIAPP led to a reduction of PFKFB3 protein expression (Fig. 3b and Supplementary Fig. 2a–b) as well as LDHA, another representative target of HIF1α.

Taken together, these results support the hypothesis that the HIF1α/PFKFB3 stress/repair pathway is activated by hIAPP toxicity, and is also activated in β-cells in humans with T2D. While activation of HIF1α was first described under conditions of hypoxia[24], it is now appreciated that HIF1α activation also occurs when the mitochondrial TCA cycle flux is attenuated, even in the absence of hypoxia (pseudo-hypoxic activation)[21,25]. Pseudo-hypoxic activation of HIF1α may occur due to mitochondrial dysfunction or adaptive quiescence of mitochondria to protect the mitochondrial network, as previously described in response to increased cytosolic $Ca^{2+}$ [26]. It is of interest that β-cells in T2D are characterized by a fragmented mitochondrial network, altered mitochondrial function, and disrupted $Ca^{2+}$ dynamics[4,7,11]. Therefore, we next sought to examine the impact of hIAPP toxicity on the mitochondrial network and β-cell function.

**hIAPP toxicity changes mitochondrial form and distribution**. The mitochondrial network in β-cells in T2D, as immunostained with Tom20 (Fig. 4a) was fragmented and less dense in appearance ($p < 0.01$) compared to β-cell mitochondria in ND (Fig. 4a, d), consistent with previous work[11]. To investigate the potential role of hIAPP toxicity in inducing these changes, we transduced INS 832/13 cells with hIAPP at an MOI known to induce misfolded protein stress[8]. Since the mitochondrial network and the dynamics of $Ca^{2+}$ vary through the cell cycle[27], we synchronized INS 832/13 cells at the G1/S stage of the cell cycle (0 h post-aphidicolin release), as confirmed by FACS analysis

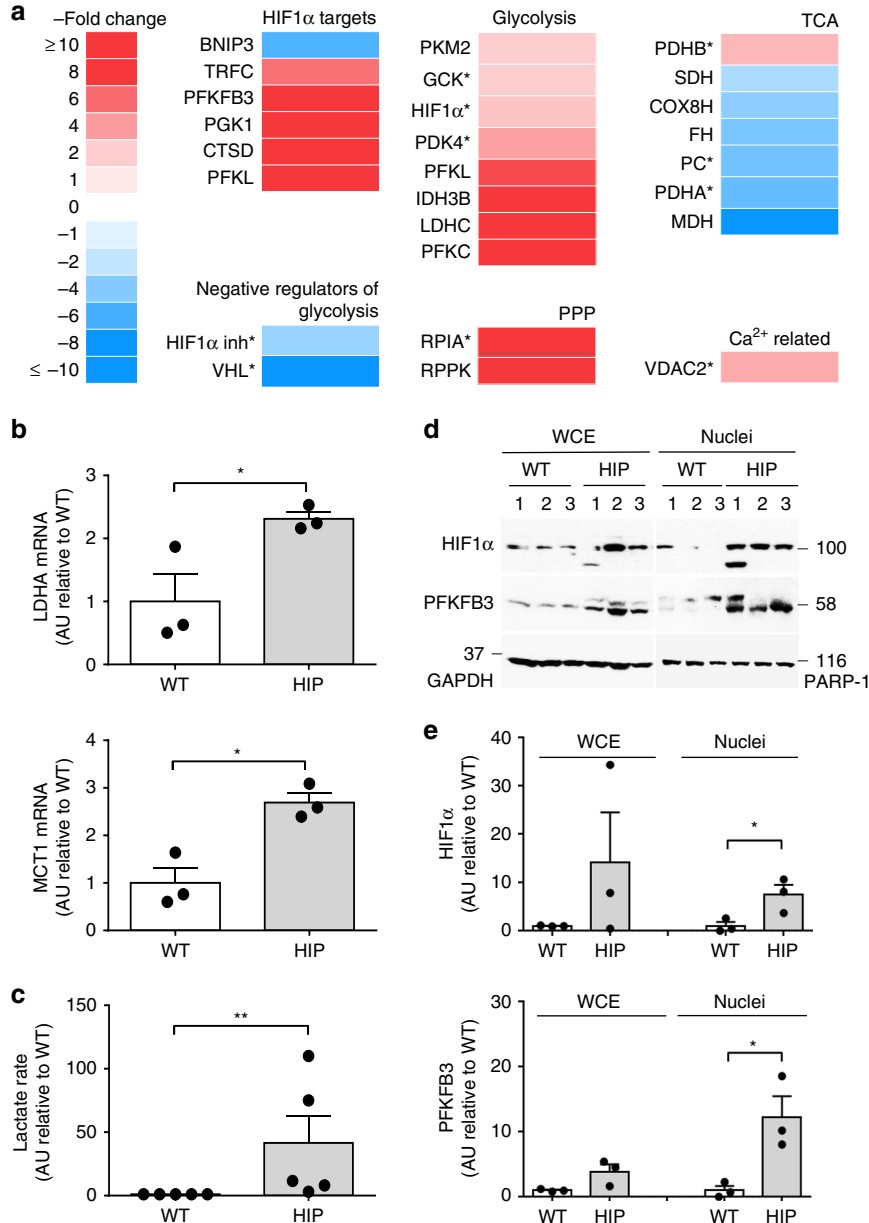

**Fig. 1** hIAPP upregulates the HIF1α/PFKFB3 stress pathway. **a** Summary of the differentially expressed genes of interest after microarray analysis[25] performed on RNA isolated from wild type (WT) and hIAPP overexpressing (HIP) rat islets (4.5 months) presented as a fold change over WT. Asterisk represents genes that were differentially expressed in one experiment. **b** LDHA and MCT1 mRNA levels in HIP versus WT as measured by qRT-PCR. **c** Lactate production rate (fold change) measured in isolated islets from HIP relative to WT. **d** Representative Western blot of PFKFB3 and HIF1α protein levels in whole cell extracts and nuclear-enriched fractions of islets from 6 months old WT (3) and HIP (3) rats. GAPDH and PARP were used as loading controls for whole cell extract (WCE) and nuclear extracts, respectively. **e** Quantification of HIF1α (upper panel) and PFKFB3 (lower panel) in cytoplasmic and nuclear fractions. Data are presented as mean ± SEM, $n = 3$ independent experiments for (**b**) and (**d**), and $n = 4$ independent experiments for (**c**). Statistical significance was analyzed by Student $t$-test (*$p < 0.05$, **$p < 0.01$)

(Supplementary Fig. 3c, d)[28]. hIAPP toxicity induced fragmentation of mitochondria (Fig. 4b, c and S4a, b) in the absence of apoptosis as confirmed by flow cytometry (Supplementary Fig. 3c). While the mitochondrial network of control cells (CTRL) was extensively reticular and tubular throughout the cytoplasm (~70% of the cells), the network was fragmented in cells overexpressing hIAPP (~50% of the cells), and displayed a perinuclear distribution (Fig. 4b, c and Supplementary Fig. 4b). Therefore, we concluded that, in common with other amyloidogenic proteins, hIAPP toxicity induces fragmentation and a perinuclear distribution of the mitochondrial network.

The mitochondrial network is continuously remodeled and repaired through regulated fission, fusion, and mitophagy[29]. To establish whether the altered mitochondrial network under conditions of hIAPP toxicity is a regulated adaptive response, we next investigated the effect of hIAPP toxicity on regulators of mitochondrial fission (dynamin related protein 1 (Drp1)) and fusion (mitofusins 1/2 (MFN1/2) and optical atrophy related 1 (Opa1)).

Immunoblotting of whole cell extracts from hIAPP transduced INS 832/13 cells revealed reduced MFN2 protein levels ($n = 3$, Supplementary Fig. 4a) but no change in Drp1 expression

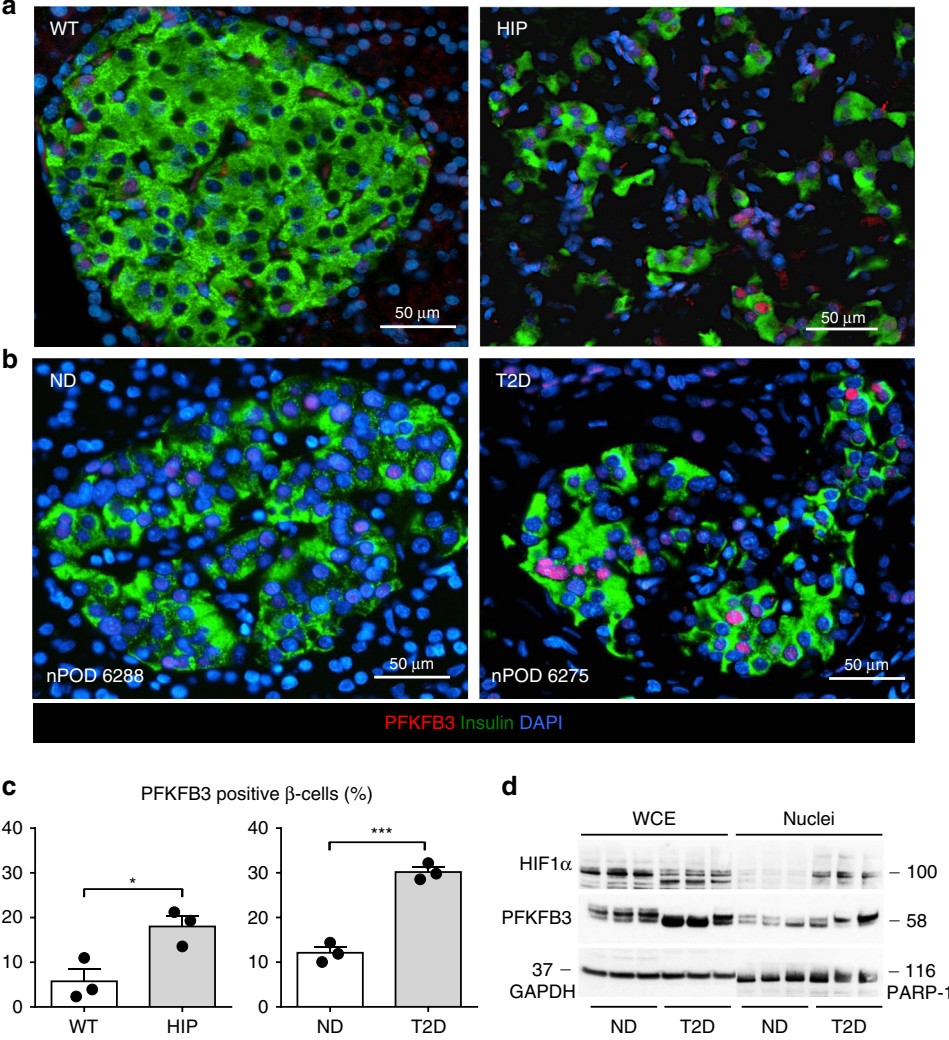

**Fig. 2** HIF1α/PFKFB3 is upregulated in β-cells of HIP rats and humans with type 2 diabetes. Representative immunofluorescence images of islets from **a** WT and HIP rats and **b** non-diabetic (ND) and T2D patients stained for PFKFB3 (red), insulin (green), and nuclei (blue). **c** Frequency of PFKFB3 positive β-cells in HIP versus WT rats (left panel) and T2D versus ND patients (right panel). **d** Representative Western blot of PFKFB3 and HIF1α levels in nuclear-enriched- and whole cell extracts from non-diabetic (ND) and T2D donor islets. Data are presented as mean ± SEM, n = 3 independent biological samples for each group. Statistical significance was analyzed by Student t-test (*$p < 0.05$, ***$p < 0.001$)

compared to controls or cells transduced with non-amyloidogenic rodent IAPP (rIAPP), implying a MFN2-dependent mechanism for mitochondrial network fragmentation (Supplementary Fig. 4a). This was supported by the finding that overexpression of the dominant negative Drp1 mutant *K48A* failed to protect against hIAPP toxicity induced mitochondrial network fragmentation (Supplementary Fig. 4b).

In conclusion, hIAPP toxicity induces an adaptive perinuclear distribution and fragmentation of the mitochondrial network mediated by decreased mitochondrial fusion, in common with other adaptive states that favor high glycolysis over oxidative phosphorylation[30–32]. We next sought to establish the impact of this change in mitochondrial network morphology on mitochondrial function.

**hIAPP toxicity induces changes in mitochondrial function.** To determine whether the altered mitochondrial network was associated with changes in mitochondrial function, we measured the cellular oxygen consumption rate (OCR) and mitochondrial membrane potential in the presence and absence of hIAPP

toxicity. We measured OCR in islets from 5–6-month old pre-diabetic HIP rats versus those from WT. There was a 30% decrease in OCR in response to 20 mM glucose in HIP rat islets compared to WT ($p < 0.01$) (Fig. 5a, b) and an increased extra-cellular acidification rate (ECAR) at both basal and stimulated glucose conditions suggesting increased acidification of the medium of HIP rat islets caused by lactate production (Supplementary Fig. 5a, b). These findings are consistent with the diversion of glycolysis to lactate production rather than pyruvate oxidation through the TCA cycle under conditions of hIAPP toxicity. Next, we investigated if the diversion of glycolysis from mitochondrial oxidative phosphorylation was secondary to a loss of mitochondrial membrane potential, or if the membrane potential was preserved despite hIAPP toxicity in a manner that would further support adaptive protective changes rather than direct mitochondrial membrane depolarization by toxic hIAPP oligomers. INS 832/13 cells expressing hIAPP or rIAPP were synchronized at G1/S (0 h post-release from aphidicolin block) and then treated with tetramethylrhodamine ethyl ester (TMRE) after which we performed flow cytometry analysis. Cells treated with carbonyl cyanide-4(trifluoromethoxy) phenylhydrazone

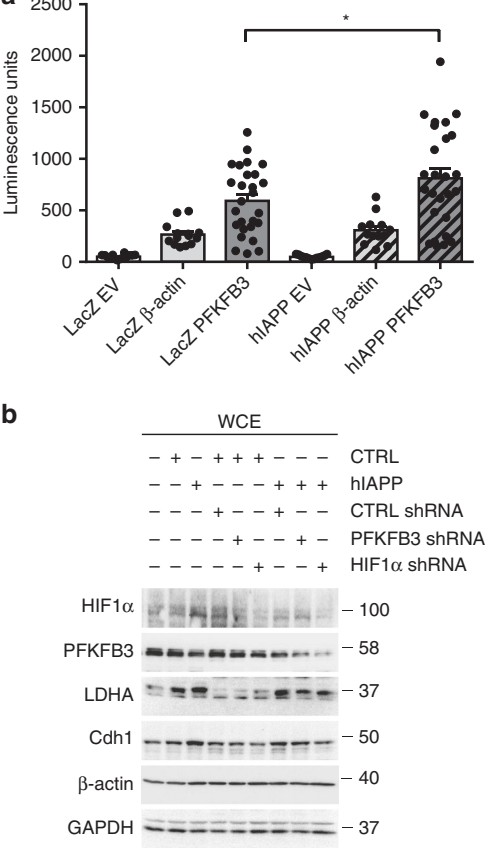

**Fig. 3** HIF1α drives the expression of PFKFB3 in β-cells. **a** Luciferase assay showing the activation of PFKFB3 promoter containing 2 hypoxia elements. INS 832/13 overexpressing LacZ or hIAPP were transfected for 36 h with plasmid vectors containing: RenSP luciferase gene without a promoter (empty vector—EV) measuring the background signal or housekeeping gene promoter driving the expression of RenSP luciferase gene (β-actin) or PFKFB3 promoter with hypoxia elements driving the expression of RenSP luciferase gene (PFKFB3). **b** Representative Western blot of HIF1α, PFKFB3, LDHA, Cdh1 in whole cell extracts of INS 832/13 overexpressing LacZ (CTRL) or hIAPP, silenced or not with short hairpin RNA for PFKFB3 (PFKFB3 shRNA) and HIF1α (HIF1α shRNA) for 36 h. CTRL shRNA is non-target shRNA control vector. β-actin and GAPDH were used as loading control. Data are presented as mean ± SEM, $n = 9$ independent biological samples (in (a)). Statistical significance was analyzed by unpaired t-test ($^*p < 0.05$).

(FCCP), a known proton ionophore, served as a positive control for the loss of mitochondrial membrane potential (Fig. 5c). The overlay of TMRE fluorescence profiles revealed no loss of mitochondrial membrane potential by hIAPP toxicity (Fig. 5d). However, inhibition of mitochondrial ATPase activity with oligomycin (5 mM) or inhibition of glycolysis with 2-deoxy-D-glucose (2-DOG, 2 mM) decreased the number of viable cells in the hIAPP group, indicating increased vulnerability to cell death upon additional toxic stimuli (Fig. 5e, f).

In conclusion, the mitochondrial network in β-cells adapts both morphologically and functionally to hIAPP toxicity, consistent with changes reported both in islets in T2D as well as neurons impacted by protein misfolding and toxic oligomers[11,12]. Given that hIAPP toxicity activates the HIF1α/PFKFB3 metabolic stress response and induces adaptive changes in mitochondrial morphology and function, we next sought to establish the metabolic consequences of these changes for β-cells.

**hIAPP increases glycolysis and pentose phosphate pathways.** To evaluate the impact of hIAPP toxicity on metabolism, we performed an unbiased metabolomics analysis in INS 832/13 cells overexpressing hIAPP versus rIAPP. Changes in the metabolite composition induced by hIAPP included decreased levels of fructose 1,6-biphosphate (F16BP), glycerol-3-phosphate (G3P), phosphoenolpyruvate (PEP), and increased levels of the glycolytic end products lactate and alanine, demonstrating enhanced glycolytic flux (Supplementary Fig. 6a, b), consistent with the increased LDHA and MCT1 gene expression we observed (Fig. 1b). Interestingly, while α-ketoglutarate (α-KG) was reduced in cells overexpressing hIAPP, its intermediate 2-hydroxyglutarate (2-HG), which may divert glucose from entry into the TCA cycle[33], showed a trend to be increased (Supplementary Fig. 6a). Reduced α-KG but maintained citrate levels are indicative of α-KG reductive carboxylation to citrate, compensating for a decreased PDH branch activity of TCA in metabolic states that mimic hypoxia[34]. Furthermore, the GSH/GSSG ratio was decreased, indicating redox stress in cells transduced with hIAPP (Supplementary Fig. 6a).

To investigate the impact of hIAPP on glucose utilization, we compared the mass isotopologue distribution (MID) of TCA intermediates using [U-$^{13}$C$_6$]-labeled glucose in INS 832/13 cells transduced with hIAPP versus rIAPP expressing adenoviruses. The intracellular M6 fraction of glucose was near 100%, demonstrating a high efficiency of labeling of glucose and its downstream metabolites (Supplementary Fig. 6b). hIAPP did not alter the glucose labeling pattern as there was no difference in the M6 fraction compared to control rIAPP overexpressing cells (Supplementary Fig. 6b). Linkage between glycolysis and the TCA cycle was disrupted by hIAPP, as demonstrated by a relative decrease in the flux of pyruvate to different glucose-derived metabolites (with M0—indicating no labeled carbons, and M1–M6—one to more labeled carbons) (Supplementary Fig. 6b). The M2 fractions of fumarate (Fum) and α-ketoglutarate (α-KG) derived from the conversion of Pyr to acetyl-CoA after the first round of the TCA cycle, were decreased in hIAPP compared to rIAPP overexpressing cells. The M3 fraction of aspartate (Asp), surrogate marker for oxaloacetate (OAA), fumarate (Fum), and malate (Mal) were also reduced, suggesting decreased pyruvate anaplerosis in hIAPP compared to rIAPP overexpressing cells (Supplementary Fig. 6b). In addition, M5 citrate was lower, suggesting a significant decrement in the use of OAA in consecutive rounds of the TCA cycle in the hIAPP cells (Supplementary Fig. 6b). Interestingly, the M0 (no labeled carbons) fraction of most TCA metabolites was increased in hIAPP cells, suggesting that another source of energy besides glucose was metabolized through the TCA cycle. Whereas the key pentose phosphate pathway (PPP) metabolite, glucose-6-phosphate/fructose-6-phosphate (G6P-F6P), was reduced (Supplementary Fig. 6a), oxidized glutathione, inositol monophosphate (IMP), and the M5 isotopologues of ADP, ATP, UDP, and UTP nucleotides were higher in hIAPP overexpressing cells, indicating an increased contribution of de novo purine and pyrimidine synthesis via the PPP, suggesting activation of a DNA damage-repair pathways (Supplementary Fig. 7a, b).

Taken together, these results indicate that hIAPP reduces the oxidation of glucose through the TCA cycle and pyruvate anaplerosis in β-cells while promoting the synthesis of nucleotides (Supplementary Fig. 7a). These findings, along with the accumulation of succinate and 2-hydroxyglutarate, are all in line with HIF1α-dependent molecular pathways[25,35]. Moreover, our findings in β-cells exposed to hIAPP toxicity resemble those previously reported in islets of patients with T2D[18] as well as neurons exposed to toxic oligomers from amyloidogenic proteins: increased aerobic glycolysis and lactate production[14]

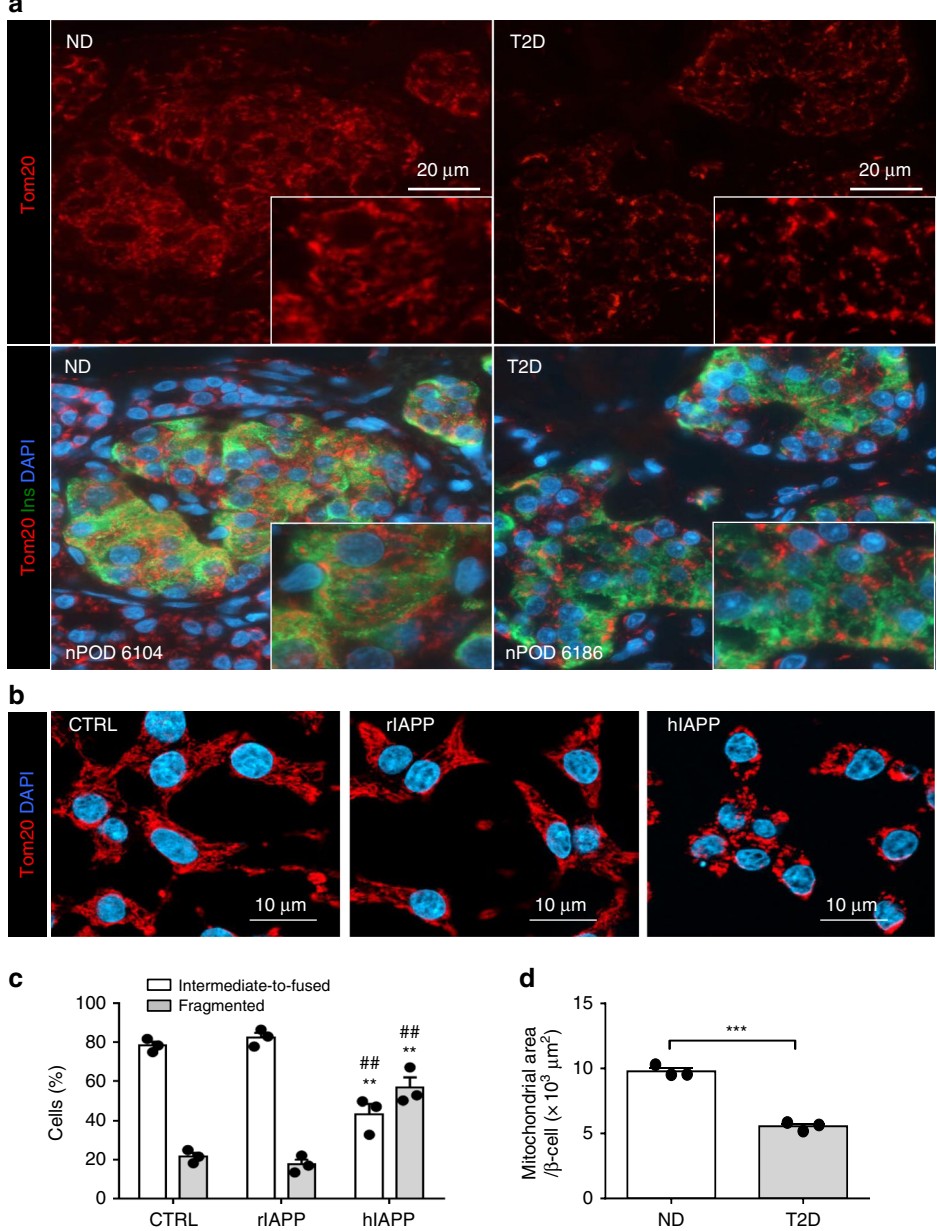

**Fig. 4** β-cell mitochondrial fragmentation. **a** Representative immunofluorescence images of islets from non-diabetic (ND) and T2D patients stained for Tom20 (mitochondria, red), insulin (green), and DAPI (nuclei, blue). **b** Representative images of INS 832/13 stained for Tom20 (mitochondria, red) and DAPI (nuclei, blue). Cells were cultured in RPMI medium, synchronized at G1/S of the cell cycle and transduced for 36 h with adenoviral vectors expressing LacZ (CTRL) or rodent IAPP (rIAPP) or human IAPP (hIAPP). **c** Quantification of mitochondrial morphology in G1/S enriched INS 832/13 cells after indicated treatments to overt fragmented or overt intermediate-to-fused mitochondria. **d** Quantification of mitochondrial area per β-cell in ND and T2D subjects. Data are presented as mean ± SEM, $n = 3$ independent experiments for each group. Statistical significance was analyzed by Student $t$-test (\*\*$p < 0.01$ relative to CTRL and ##$p < 0.01$ relative to rIAPP)

implying disengagement of glucose metabolism from mitochondrial respiration.

**PFKFB3 mediates hIAPP-induced changes in metabolism**. To establish if the metabolic reprogramming of β-cells under conditions of hIAPP toxicity is mediated by the HIF1α/PFKFB3 stress response pathway, we next investigated the effect of silencing PFKFB3 on hIAPP-induced changes in β-cell metabolism. PFKFB3 silencing restored most metabolites to their control levels (Fig. 6a). ADP/ATP and AMP/ATP ratios were normalized, as were lactate and palmitate levels and metabolites of the homocysteine pathway (Fig. 6a).

To investigate the impact of PFKFB3 inhibition on the metabolic fate of glucose, we analyzed the MID of TCA metabolites derived from culturing INS 832/13 cells transduced with rIAPP or hIAPP, in the presence or absence of PFKFB3, with [U-$^{13}$C$_6$]-labeled glucose. PFKFB3 silencing led to a decrease of pyruvate anaplerosis via OAA as demonstrated by a relative reduction of the M3 fractions of Mal, Asp, and Fum (Fig. 6b and Supplementary Fig. 8). However, the conversion of acetyl-CoA to Cit, α-KG, and Fum was increased in β-cells transduced with hIAPP and silenced for PFKFB3 as demonstrated by the increased abundance of the M2 isotopologue fraction of these metabolites and implying re-engagement of glucose metabolism to the TCA

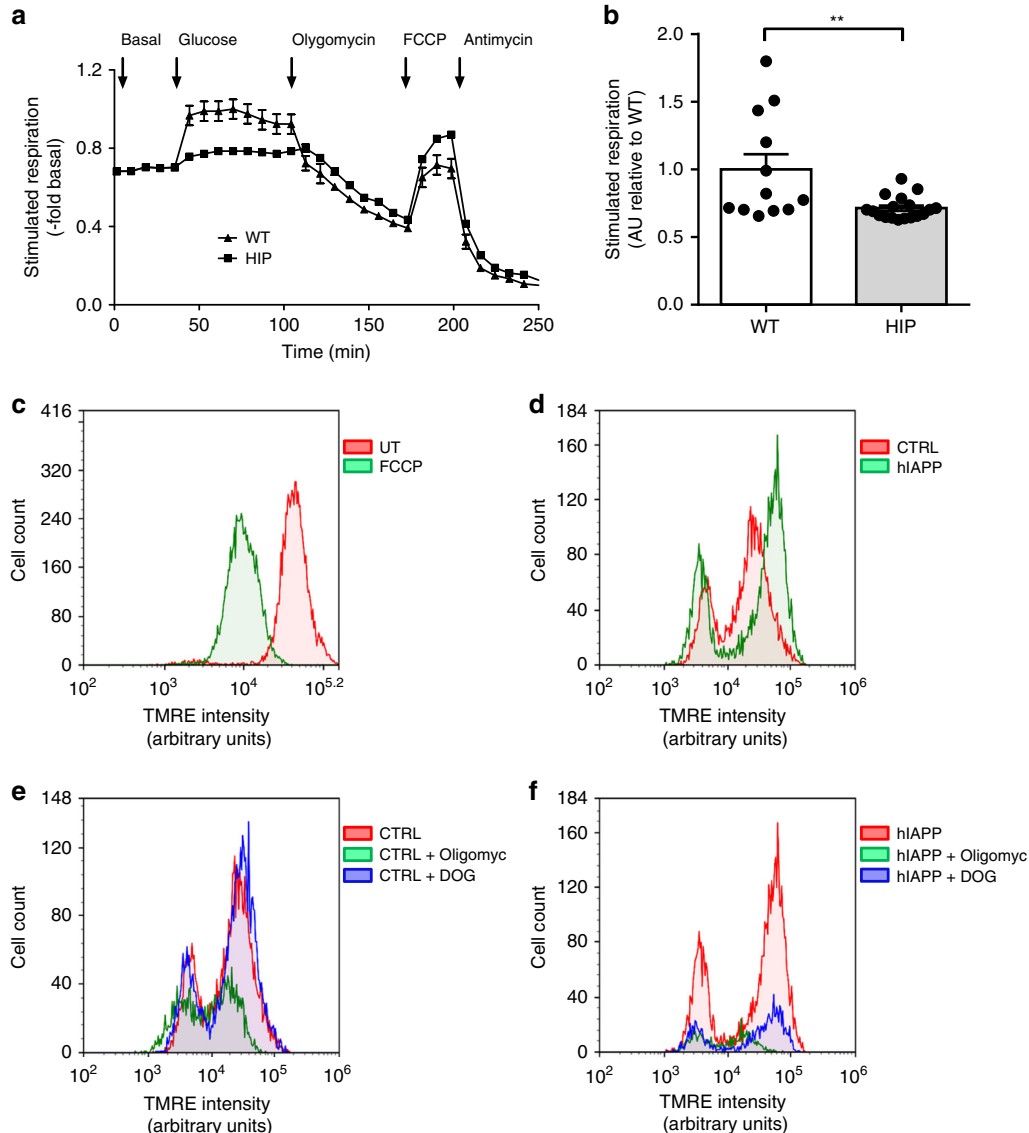

**Fig. 5** Mitochondrial respiration is decreased by hIAPP. **a** Oxygen consumption rate (OCR) measurements were obtained over time (min) using an extracellular flux analyzer (Seahorse Bioscience). Bioenergetic parameters were obtained by adding glucose (20 mM) to derive OCR under stimulated conditions, the ATP synthase inhibitor Oligomycin A to derive ATP-linked OCR, FCCP to uncouple the mitochondria for maximal OCR and antimycin. Bioenergetic profile of isolated islets from 4–6 months old HIP and WT rats from one experiment. **b** Quantification of OCR is presented as a fold change of stimulated respiration versus WT. Mitochondrial membrane potential was measured by flow cytometry after labeling with TMRE dye in asynchronous cells in the presence or absence of the mitochondrial uncoupling agent FCCP (**c**), (LacZ) CTRL versus hIAPP overexpressing cells synchronized in G1/S of the cell cycle (**d**), CTRL (**e**) or (**f**) hIAPP overexpressing cells in G1/S in the presence or absence of oligomycin (Oligomyc) or 2-deoxy-glucose (DOG). Data are presented as mean ± SEM, $n = 3$ independent experiments for each group. Statistical significance was analyzed by Student $t$-test (**$p < 0.01$)

cycle via PDH, at least after the first round of the TCA cycle (Fig. 6b). The lactate levels were reduced after PFKFB3 inhibition (Fig. 6c) confirming that the PFKFB3 suppression reduced the flux through glycolysis, as expected from our hypothesis.

Several isotopologue fractions of the PPP intermediates, G6P-F6P (M2–M5), ribulose-5-phosphate (R5P) (M3), and sedoheptulose-7-phosphate (S7P) (M1–M6), were increased in hIAPP transduced cells when PFKFB3 was silenced, indicating the oxidation of glucose in this pathway was upregulated (Supplementary Fig. 8). In conclusion, inhibition of PFKFB3 promotes a partial re-engagement of glucose entry into the mitochondrial TCA cycle via acetyl-CoA in β-cells overexpressing hIAPP, while permitting further its utilization in the PPP.

Having established that PFKFB3 implements many of the changes in metabolism induced by hIAPP toxicity, we next

investigated the role of PFKFB3 in mediating the hIAPP toxicity-induced changes in the mitochondrial network. The re-engagement of glycolysis to the TCA cycle due to PFKFB3 silencing in β-cells exposed to hIAPP toxicity restored the fused form of the mitochondrial network (Fig. 7a, b). The fragmentation and perinuclear clustering of the mitochondrial network observed in cells exposed to toxicity induced by amyloidogenic proteins has been attributed to a protective adaptation to aberrant cytosolic Ca$^{2+}$ [26]. We next investigated β-cell Ca$^{2+}$ levels under conditions of hIAPP toxicity and after PFKFB3 silencing.

**Activated PFKFB3 contributes to aberrant β-cell Ca$^{2+}$ dynamics.** The exocytosis of insulin granules is triggered by Ca$^{2+}$ influx through voltage-gated Ca$^{2+}$ channels into the cytosol of

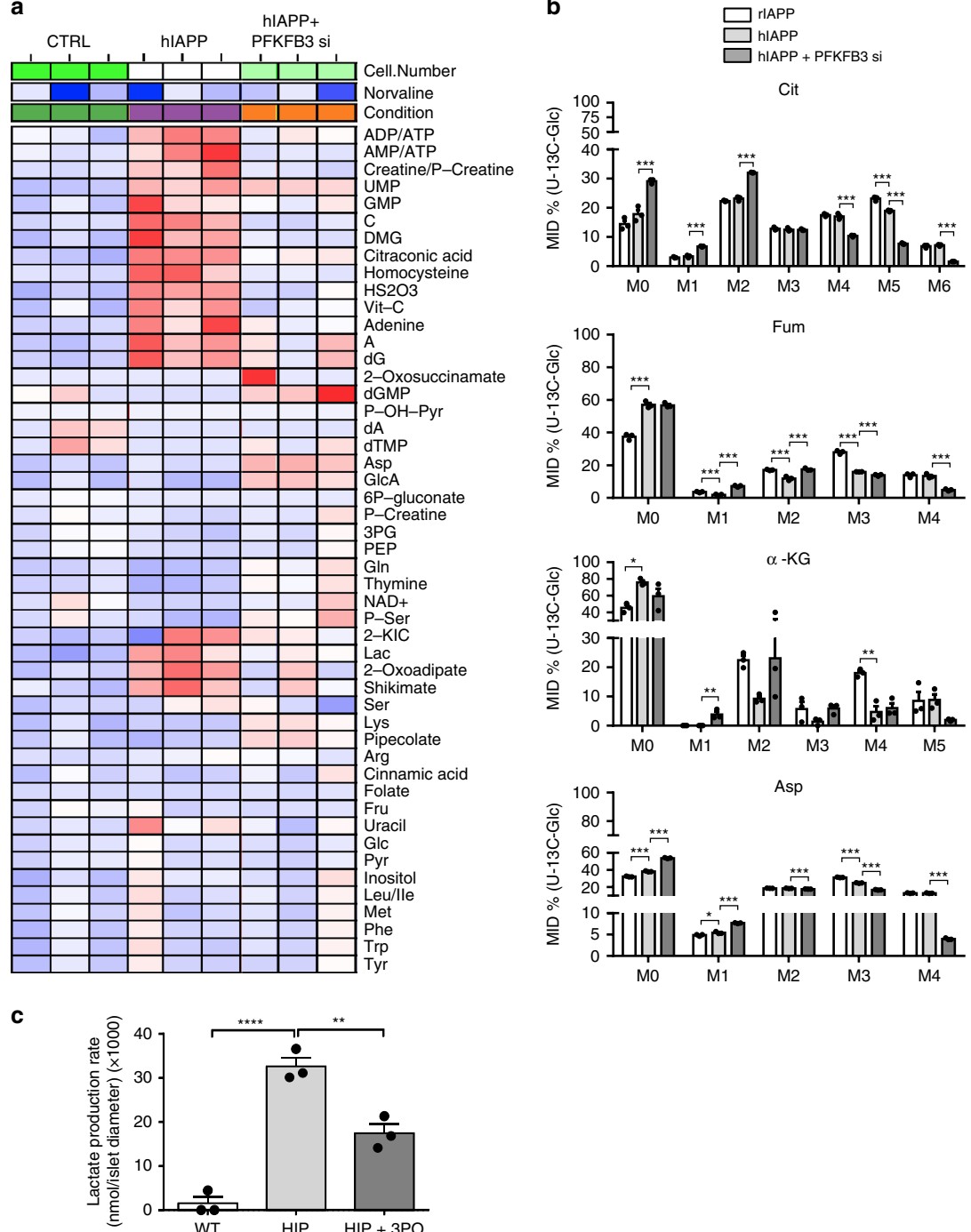

**Fig. 6** Silencing of PFKFB3 restores the β-cell metabolome. **a** Heatmap of relative metabolite composition in LacZ (CTRL) and hIAPP overexpressing INS 832/13 cells in the presence or absence of PFKFB3 siRNA. **b** Mass isotopomer distribution (MID) of the TCA metabolic intermediates from cultured INS 832/13 cells with [U-$^{13}$C$_6$] labeled glucose. Citrate—Cit; Fumarate—Fum; α-ketoglutarate—α-KG; Aspartate—Asp. **c** Lactate production rate of islets from WT and HIP rats treated or not with 3PO for 24 h. Data are presented as mean ± SEM, $n = 3$ independent biological samples for each group. Statistical significance was analyzed by one-way ANOVA test with Tukey's post-test (*$p < 0.05$, **$p < 0.01$, ***$p < 0.001$)

the β-cell[36]. These channels are activated (opened) by depolarization of the β-cell membrane as a consequence of the closure of K(ATP) channels by a rise in ATP/ADP. Cellular toxicity induced by amyloidogenic proteins is linked with sustained aberrant Ca$^{2+}$ signaling, and this is evident in affected cells, including β-cells in T2D, by calpain hyperactivation[7,8].

To investigate the impact of hIAPP toxicity on β-cell Ca$^{2+}$, we first investigated islets isolated from hIAPP transgenic mice in 11 mM glucose (submaximal). As expected, cytosolic Ca$^{2+}$ was

elevated in islets of hIAPP transgenic mice compared to those from rIAPP transgenic controls (Fig. 7c).

In order to further probe the impact of hIAPP toxicity on Ca$^{2+}$ dynamics, we next measured [Ca$^{2+}$] in the cytosol, as well as the ER, and mitochondria under basal (2.8 mM) and stimulated (16.8 mM) glucose in INS 832/13 cells exposed to hIAPP toxicity versus controls. In control cells, raising glucose to 16.8 mM increased mean cytosolic, ER, and mitochondrial [Ca$^{2+}$] (Fig. 8a–c and Supplementary Fig. 9a–f) as previously reported[37,38] and the

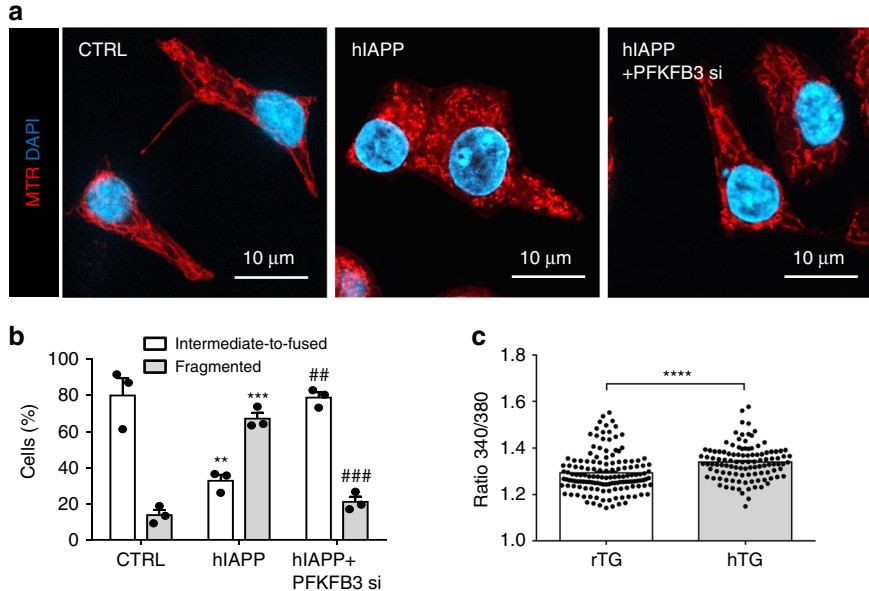

**Fig. 7** PFKFB3 silencing restores the mitochondrial network in hIAPP toxicity. **a** Representative immunofluorescence images of INS 832/13 stained with Mitotracker red (MTR) (mitochondria, red), and DAPI (nuclei, blue). Cells were serum-starved for 56 h and transduced with adenoviral vectors expressing LacZ or hIAPP, silenced or not for PFKFB3 for 36 h. **b** Quantification of mitochondrial morphology in serum-starved INS 832/13 cells after indicated treatments to overt intermediate-to-fused or overt fragmented. **c** Cytosolic calcium concentration measured in isolated islets from rodent IAPP (rTG) and human IAPP (hTG) transgenic mice. Data are presented as mean ± SEM, $n = 3$ independent experiments for each group. Statistical significance was analyzed by Student $t$-test ($*p < 0.05$, $**p < 0.01$, $***p < 0.001$).

addition of the SERCA blocker cyclopiazonic acid (CPA) or the mitochondrial inhibitor sodium azide (NaN$_3$) decreased ER or mitochondrial [Ca$^{2+}$], respectively. Individual Ca$^{2+}$ traces detailing the Ca$^{2+}$ dynamics of each compartment for the different experimental conditions examined are shown in Supplementary Fig. 9.

In INS 832/13 cells under conditions of hIAPP toxicity, cytosolic [Ca$^{2+}$] was elevated compared to controls ($p < 0.05$ versus LacZ_CTRL siRNA) in 2.8 mM glucose and increased further compared to controls when challenged with 16.8 mM glucose (Fig. 8a and Supplementary Fig. 9a, d). ER and mitochondrial [Ca$^{2+}$] were higher than in controls ($p < 0.05$) under basal conditions at 2.8 mM glucose but were relatively unresponsive to an increase in glucose to 16.8 mM (Fig. 8b, c and Supplementary Fig. 9b, c, e, f). However, both ER and mitochondria were actively sequestering Ca$^{2+}$ despite hIAPP toxicity, since pharmacologically inhibiting ER or mitochondrial Ca$^{2+}$ uptake led to store depletion, inferring maximal levels of stored [Ca$^{2+}$]. Importantly, silencing PFKFB3 under conditions of hIAPP toxicity in INS 832/13 cells partially restored the levels and dynamics of Ca$^{2+}$ in all three compartments (i.e., cytosolic, ER, and mitochondrial) both under basal and glucose stimulatory conditions (Fig. 8a–c). These data imply that the increased basal cytosolic, ER, and mitochondrial [Ca$^{2+}$] and the diminished glucose responsiveness observed during hIAPP toxicity might be at least in part due to the sustained closure of K(ATP) channels by high basal glycolytic flux induced by PFKFB3[39,40]. Of interest, inhibition of the K(ATP) channel has been reported to be protective against β-cell loss in several models of diabetes[41,42].

On the one hand, these data could imply that inhibition of PFKFB3-mediated metabolic remodeling might be protective against hIAPP cytotoxicity. On the other hand, given that the HIF1α/PFKFB3 signaling pathway is considered as protective against cytotoxicity, inhibition of this signaling pathway might be deleterious to cell survival under conditions of hIAPP toxicity. To address this, we next examined the impact of

inhibition of HIF1α or PFKFB3 signaling on β-cell survival under conditions of hIAPP toxicity.

**HIF1α/PFKFB3 signaling attenuates hIAPP stress-induced apoptosis**. In order to establish if hIAPP-induced HIF1α signaling is indeed protective against cell death, we exposed INS 832/13 cells transduced with hIAPP to a well characterized and specific HIF1α inhibitor[43]. Under these conditions, hIAPP-induced β-cell apoptosis was enhanced ($p < 0.05$) as affirmed by western blotting (cleaved caspase-3), TUNEL immunostaining, and FACS analysis that revealed an increased proportion of sub-G1 cells (Fig. 9 and Supplementary Fig. 10).

We undertook comparable experiments in INS 832/13 cells transduced with hIAPP to establish if PFKFB3 is protective against hIAPP-induced β-cell death by using validated PFKFB3 siRNA. β-cell apoptosis was also increased by suppression of PFKFB3 ($p < 0.05$) as evaluated by activation of cleaved caspase-3, TUNEL immunostaining, and by FACS analysis (Fig. 9 and Supplementary Fig. 10).

In conclusion, the HIF1α/PFKFB3 signaling pathway is protective against hIAPP-induced β-cell apoptosis.

## Discussion

Protein misfolding and the toxicity of amyloidogenic proteins reflect a mismatch between the rate of synthesis of the protein versus the capacity of the cell to properly fold, traffic, and clear the misfolded proteins. IAPP expression increases with insulin resistance[44], and is a major risk factor for T2D, and as well, the capacity to clear misfolded proteins declines with age[45] in long-lived cells such as β-cells and neurons. Moreover, the toxicity of misfolded protein further compromises the autophagy and ubiquitin proteasome pathways' capacity to clear misfolded proteins[46,47]. Therefore, once β-cell IAPP misfolded protein toxicity develops, it is likely to be sustained unless there is a

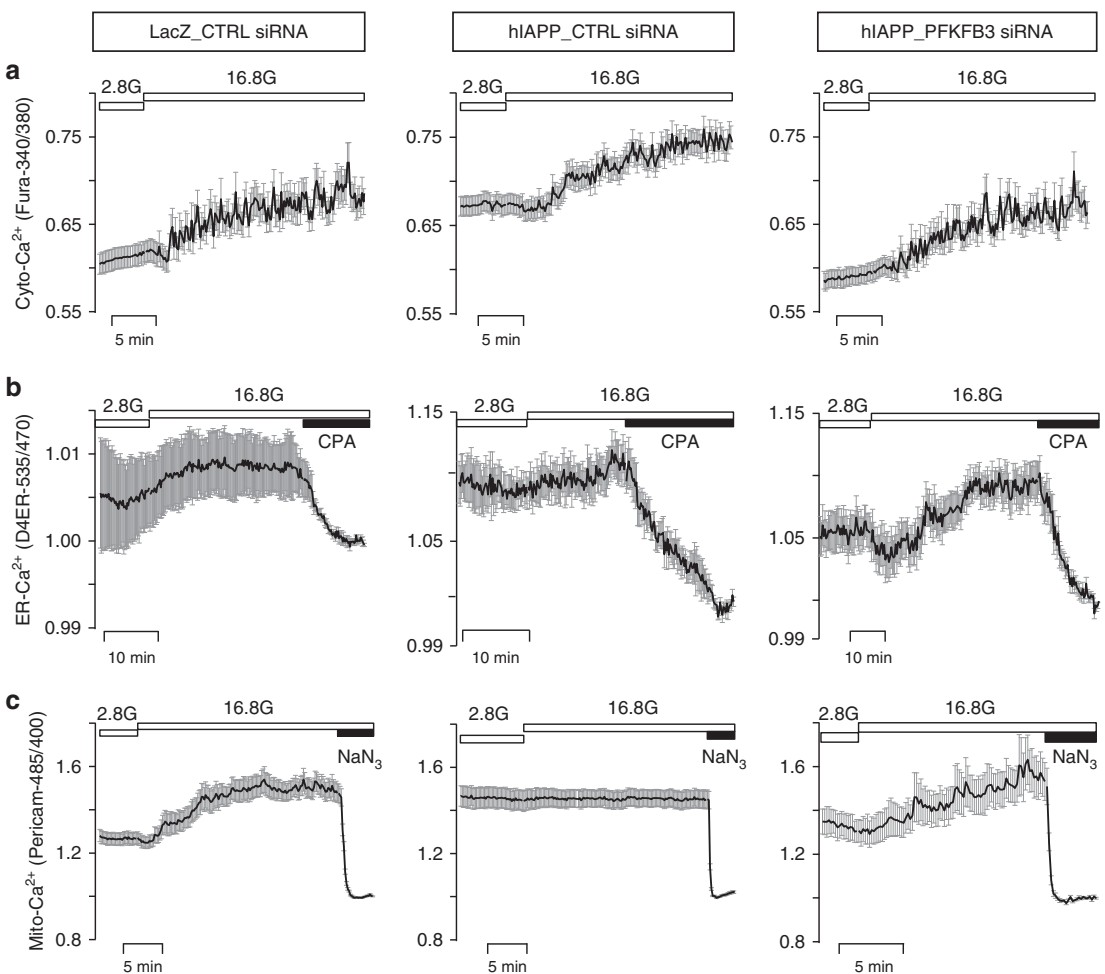

**Fig. 8** PFKFB3 silencing restores $Ca^{2+}$ dynamics of cytosolic, ER and mitochondrial compartment. Average traces ± SEM of **a** cytosolic, **b** endoplasmic reticulum (ER), and **c** mitochondrial calcium in INS 832/13 cells overexpressing LacZ (LacZ_CTRL siRNA) or hIAPP in the presence or absence of PFKFB3 siRNA (hIAPP_PFKFB3 siRNA and hIAPP_CTRL siRNA, respectively) measured at low (2.8 mM, 2.8 G) and high (16.8 mM, 16.8 G) glucose. Cyclopiazonic acid (CPA) and sodium azide ($NaN_3$) have been added after measurement of $[Ca^{2+}]$ in high glucose (16.8 mM) to establish the $[Ca^{2+}]$ content in ER and mitochondria, respectively. Average $Ca^{2+}$ traces are presented from one of 3 ($n = 3$) independent experiments. The vertical axis in LacZ_CTRL siRNA for D4ER 535/470 ratio has been shown within 0.99–1.015 AU to indicate the increment in average traces upon exposure to high (16.8 mM) glucose. $Ca^{2+}$ traces shown for ER and mitochondria were first normalized to their respective nadirs, obtained in the presence of CPA or $NaN_3$

marked increase in insulin sensitivity, such as following childbirth.

In this study, we discovered that changes in the mitochondrial network and metabolism induced by hIAPP toxicity recapitulate the phenotype of β-cells in T2D[11,48] as well as of neurons in neurodegenerative diseases[13,14,49,50]. We were initially drawn to these potential protective effects of altered metabolism given that they are reminiscent of those present in cancer cells that adaptively resist cell death. We propose that in non-replicative cells such as β-cells and neurons, in the face of chronic misfolded protein stress, toxicity is countered by the protective metabolism induced by the HIF1α/PFKFB3 metabolic stress pathway, providing an explanation for the slow rate of attrition of cells at the expense of cell dysfunction. This concept is supported by our finding of enhanced hIAPP-induced β-cell toxicity when HIF1α activation was suppressed.

The main adaptive metabolic response mediated by activation of the HIF1α/PFKFB3 pathway was the disengagement of glycolysis from the mitochondrial TCA cycle along with adaptive fragmentation of the mitochondrial network (Fig. 10). The HIF1α stress response thus provides short-term survival benefit in response to acute stress, such as a hypoxic event[51–53]. However,

given the unique dependence of β-cell function on tight engagement of glucose with the TCA cycle, this adaptive change predictably induces β-cell dysfunction with relatively high insulin secretion occurring at baseline glucose values (likely because of ATP generated by unrestrained glycolysis), but also a deficient response to glucose stimulation, both characteristics of β-cells in T2D[54].

The PFKFB3-mediated increased flux through glycolysis under conditions of hIAPP toxicity also predominately redirects pyruvate to lactate. The latter regenerates NAD for the maintenance of the redox state in mitochondria[55]. Also, under conditions of hIAPP toxicity the enhanced glucose flux through glycolysis is partitioned to a greater extent than in healthy β-cells through the PPP, typically activated under conditions of stress or during cell replication to provide nucleotide precursors needed for DNA synthesis and/or repair as well as reducing equivalents. Therefore, β-cells experiencing hIAPP-induced stress adopt a metabolic pattern that mimics the so called Warburg effect reported in cancer cells[15]. We observed increased PFKFB3 expression, mainly in the β-cell nuclei of both HIP rats and humans with T2D. PFKFB3 allosterically activates the downstream PFK1 through its product fructose 2,6 biphosphate (F2,6BP), and is also known to

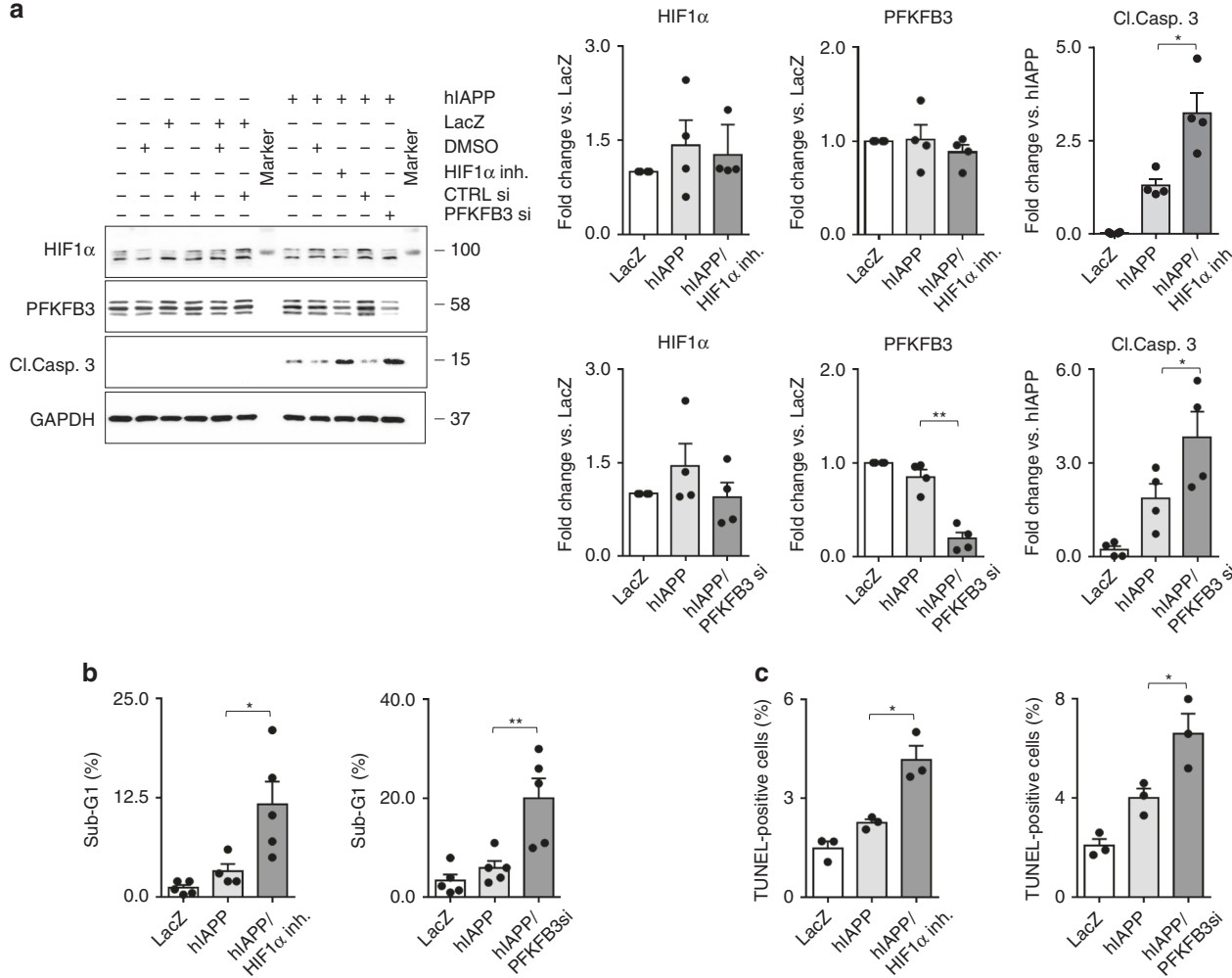

**Fig. 9** HIF1α and PFKFB3 are protective against hIAPP mediated β-cell death. **a** Representative immunoblotting (left) and quantification (right) of the whole cell extract (WCE) from asynchronous INS 832/13 cells treated with either HIF1α inhibitor KC7F2 (10 μM) or PFKFB3 siRNA (75 nM) for 56 h versus control. Cells were transduced with hIAPP adenovirus (75 MOI) or LacZ adenovirus as control (75 MOI) for last 32 h of culture. DMSO or CTRL siRNA, respectively were added to LacZ or hIAPP to equal DMSO or PFKFB3 siRNA used in experimental groups. GAPDH was used as a loading control. Data are presented as mean ± SEM, n = 4, *p < 0.05, **p < 0.01. **b** Quantification of DNA content distribution after FACS analysis of INS 832/13 cells treated as described in (**a**). Data are the mean ± SEM, n = 4, *p < 0.05, **p < 0.01. **c** The frequency of TUNEL positive INS 832/13 cells treated as described in (**a**). Data are the mean ± SEM, n = 4 in (**a**) and (**b**), and n = 3 independent experiments in (**c**). Statistical significance was analyzed by one-way ANOVA test with Tukey's post-test (*p < 0.05, **p < 0.01). See Supplementary Fig. 10 for additional supportive data

regulate cellular growth. Metabolites such as F2,6BP can pass unassisted by diffusion through the nuclear pore complex and activate PFK1 without any necessity for PFKFB3 to undergo cytoplasmic shuttling[56]. Previous studies linked PFKFB3 with signaling that promotes entry into the cell cycle[57,58] presumably to facilitate the regenerative phase of the HIF1α/PFKFB3 stress response pathway. While adult β-cells have a limited capacity to complete the cell cycle, the partial dedifferentiation of β-cells previously reported in T2D[59,60] may reflect sustained signaling for entry into, but a failure to execute, the cell cycle. Partial dedifferentiation is a regulated step in preparation for replication by differentiated cells such as β-cells[61]. The adaptive changes in metabolism and mitochondrial network induced by the HIF1α/PFKFB3 pathway in response to hIAPP toxicity are also present in β-cells in T2D and are comparable to those present in replicating β-cells[27]. Immature β-cells retain comparable metabolism presumably to permit cell replication, so it is not surprising that β-cells exposed to the sustained activation of the HIF1α/PFKFB3 pathway might be considered as adopting an immature dedifferentiated state. The current study provides a plausible mechanism for that process and establishes that it protects β-cell viability against stress at the expense of β-cell function.

The importance of mitochondrial network remodeling as a means to regulate cell metabolism was recently elegantly illustrated in T-cells[31]. Mitochondrial fragmentation seen in response to hIAPP overexpression in β-cells was associated with reduced glucose flux through the TCA cycle, as demonstrated by measurements of OCR and mass spectrometry. Mitochondria network form and disposition is regulated by alterations in the balance of network fusion and fission. The more fragmented form in response to hIAPP was mediated by a decrease in the fusion protein MFN2, mirroring the mechanism subserving adaptation to the more fragmented network form seen in response to stress in neurons[26] and activation of T-cells[62] and in the regulated adaptation of brown fat cells so as to withstand the stress of insulin resistance[63]. A more fragmented perinuclear mitochondrial network has been shown to be protective against the potentially deleterious effects of aberrantly high cytosolic Ca$^{2+}$ waves[26,64] as present in hIAPP toxicity, and following ischemic reperfusion injury in cardiomiocytes[52]. Of interest the latter are

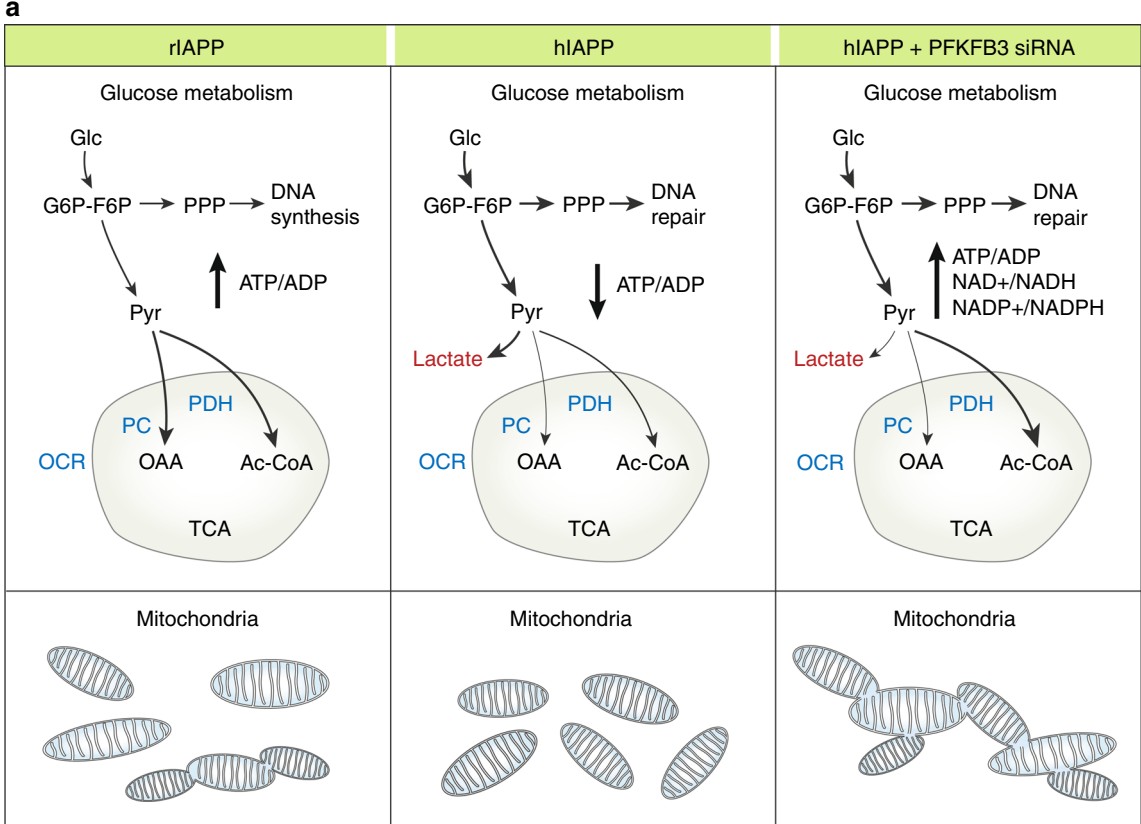

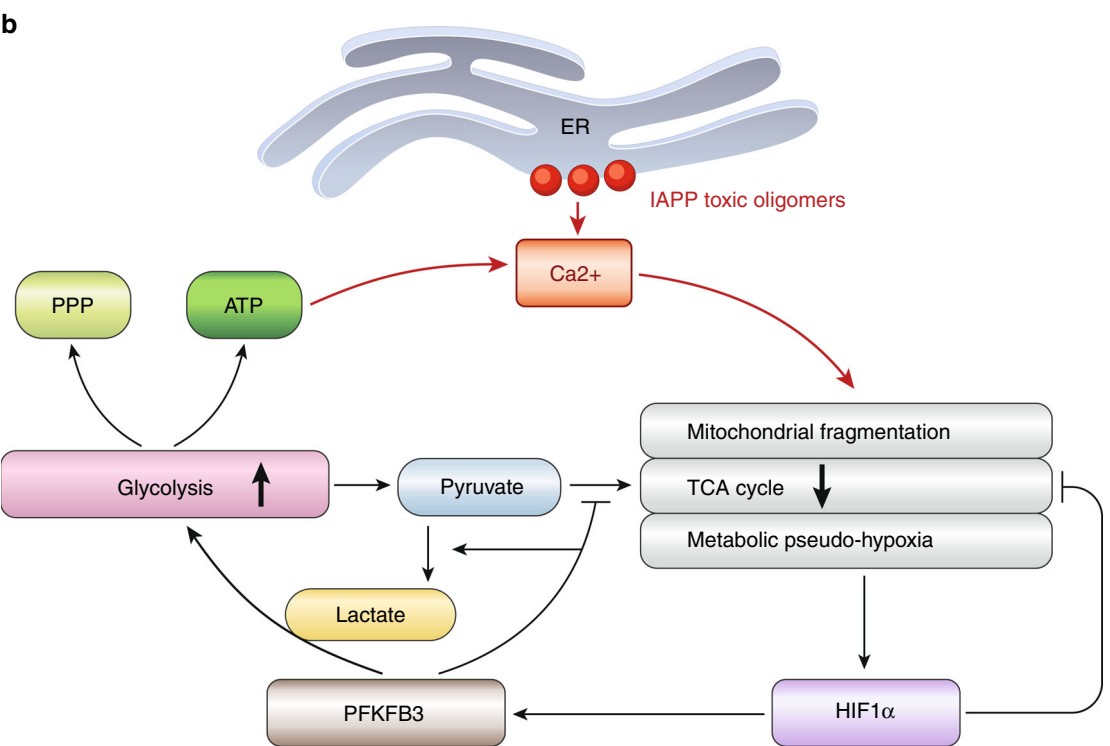

**Fig. 10** Scheme and working model. **a** Schematic of the metabolic changes in β-cells overexpressing rIAPP, hIAPP, or hIAPP silenced for PFKFB3. In comparison with INS 832/13 overexpressing rIAPP, cells transduced with hIAPP show fragmentation of the mitochondrial network and upregulation of glycolysis with lactate production. PFKFB3 siRNA partly reverts these metabolic changes in cells overexpressing hIAPP. **b** Working model illustrating hIAPP toxicity induces increased β-cell cytosolic [$Ca^{2+}$], initially most likely as a consequence of the nonselective membrane permeability at the ER. In response to high cytosolic [$Ca^{2+}$], mitochondria adopt a defensive fragmented posture leading to reduced TCA momentum and metabolic pseudo-hypoxia important for activation of HIF1α and its target PFKFB3. PFKFB3 enhances the aerobic glycolysis that diverts pyruvate into lactate and stimulates the pentose phosphate pathway to forge repair. However, glycolysis also produces ATP that may sustain the closure of the K(ATP) channels and concomitantly, $Ca^{2+}$ influx

also protected from cell death by activation of the HIF1α stress pathway[52].

Given that the initial response to hIAPP toxicity appears to be protective, the question arises: what underpins the eventual failure of these adaptive responses in β-cells, and are there therapeutic strategies that might be employed to sustain protection? Stabilization of HIF1α has been reported to occur in response to the accumulation of TCA intermediates such as fumarate, citrate, or hydroxyglutarate when TCA cycle flux declines, as occurs during short term hypoxia[21,25,35]. Under such stress conditions, the marked acceleration of glycolysis to generate ATP with enhanced DNA synthesis for DNA repair are clearly desirable attributes to promote cell survival. Moreover, upon cessation of the short-term injury, cells that survive with minimal DNA damage and are replication competent can pass through the cell cycle and contribute to tissue repair. In both β-cells and neurons undergoing misfolded protein stress there are several predictable reasons why the HIF1α/PFKFB3 stress pathway becomes chronically activated.

First, misfolded protein-induced stress is typically a characteristic of aging, occurring when protective mechanisms against proteotoxicity, including chaperone proteins, autophagy, and the proteasome, are in decline[45,65]. Therefore, the protein misfolding stress, once initiated, is typically sustained and indeed self-perpetuating. Second, both β-cells and neurons are highly dependent on the tight linkage between glycolysis and the TCA cycle[66,67]. With sustained dysregulated ATP production through a high glycolytic flux, both neurons and β-cells are vulnerable to accumulate DNA damage and undergo a process of cell identity loss or dedifferentiation[59,68]. Moreover, key aspects of cell function (for example the regulated exocytosis of insulin or neurotransmitters) in neurons and β-cells require tight regulation of subcellular $Ca^{2+}$ transients. However, with a sustained high glycolysis, harmful aberrant cytosolic $Ca^{2+}$ signaling occurs that contributes to calpain hyperactivation in both neurons and β-cells[7,8,69].

In order to understand to what extent the metabolic phenotype and/or loss of $Ca^{2+}$ homeostasis observed under misfolded protein stress by hIAPP relies on PFKFB3, we suppressed accelerated glycolysis by silencing PFKFB3. Silencing PFKFB3 suppressed hIAPP-increased glycolytic flux, restored pyruvate incorporation into TCA to generate Ac-CoA, while still permitting the adaptive increase in the PPP pathway. The latter supports not only DNA repair but also the increased formation of $NAD^+$ and $NADP^+$ for protection of β-cells against ROS to which they are particularly vulnerable[70]. Moreover, higher $NAD^+$ availability by PFKFB3 downregulation may contribute to the observed restoration of the mitochondrial network[71] in a $NAD^+$-dependent manner. Interestingly, reducing the levels of PFKFB3 and aerobic glycolysis partially restored cytosolic, ER, and mitochondrial $[Ca^{2+}]$ levels as well as organellar glucose-dependent $Ca^{2+}$ dynamics observed in controls.

In conclusion, these studies provide a potential link between the IAPP misfolded protein stress characteristic of β-cells in T2D and the seemingly disparate alterations in β-cell function known to occur in T2D. IAPP misfolded protein stress activates the conserved HIF1α/PFKFB3 injury regeneration pathway, which is also activated in β-cells in T2D. The metabolic and mitochondrial network changes in β-cells in T2D reflect the pro-survival adaptations of the HIF1α/PFKFB3 pathway. The β-cell metabolic and mitochondrial remodeling induced by the HIF1α/PFKFB3 pathway in T2D provides an explanation for the slow rate of β-cell loss at the expense of β-cell function, mirroring the time course and functional changes present in neurons impacted by protein misfolding. The adaptive changes mediated by sustained HIF1α/PFKFB3 signaling pathway likely also provide the mechanism underlying the recently described phenomenon of partial β-cell dedifferentiation in T2D. Moreover, sustained activation of the HIF1α/PFKFB3 stress response pathway and the consequent metabolic remodeling that occurs in response to misfolded protein stress provides a unifying explanation for the early loss of cell function (i.e., β-cell glucose responsiveness) but slow rate of β-cell loss in misfolded protein diseases such as Alzheimer's disease and T2D. Inhibition of the pathway without removing misfolded protein stress would likely hasten cell loss in these diseases.

## Methods

**Cell culture.** The rat insulinoma cell line INS 832/13 was provided by Dr. Christopher Newgard (Duke University, Durham, NC). INS 832/13 cells were cultured in RPMI 1640 medium supplemented with 10 mM HEPES, 1 mM sodium pyruvate, 100 IU/ml penicillin and 100 mg/ml streptomycin (Invitrogen, Carlsbad, CA, USA), 10% heat-inactivated fetal calf serum (FCS) (Gemini Bio-Products, West Sacramento, CA, USA), and 50 μM β-mercaptoethanol (Sigma, St. Louis, MO, USA) at 37 °C in a humidified 5% $CO_2$ atmosphere. For studies to investigate the role of HIF1α and PFKFB3 on protection against cell death, we made use of the HIF1α inhibitor, KC7F2 (Selleckchem). Asynchronous INS 832/13 cells were seeded at cell density of $0.8 \times 10^6$ per/well in 6-well plates. Next day cells were treated with the HIF1α inhibitor (10 μM in DMSO) or DMSO vehicle only, for 24 h prior transduction of hIAPP or LacZ adenoviruses at 75 MOI or PFKFB3 siRNA (75 nM, see the sequence below in the section "Small interfering RNA") for 32 h. Aliquots of cells were collected for protein expression- by immunoblotting and flow cytometry analysis, while cells seeded on the coverslips were fixed with 4% paraformaldehyde solution in PBS at room temperature for 10 min before being subjected to TUNEL assay according to manufacturer's instructions (Roche).

**Cell cycle synchronization.** INS 832/13 cells were plated in culture medium with 10% FCS for 24 h. Medium was then replaced with fresh medium containing 0.1% FCS for 56 h to allow cells to reach the G0 out-of-cycle state. Synchronization of cells in G1/S, S, and G2/M stages of cell cycle was carried out as follows: after 24 h in medium containing 10% FCS, cells were maintained in culture medium + 0.1% FCS for 56 h. Medium was replaced with fresh medium + 10% FCS and, 12 h later, aphidicolin was added. After 12 h treatment with aphidicolin (Sigma A0781, St. Louis, MO, USA), the medium was replaced with medium containing 10% FCS w/o aphidicolin and the cells were collected at 0, 4, 12 h after aphidicolin release. Cell cycle distribution was determined based on flow cytometry profiling of DNA content.

**Cell cycle distribution and sub-G1 analysis by flow cytometry.** Cells were trypsinized, washed with ice-cold PBS, and fixed in 80% methanol at −20 °C for at least 2 h. Cells were stained with propidium iodide (50 μg/ml) in the presence of RNase A (50 μg/ml) in PBS for 30 min at 37 °C after methanol was removed by centrifugation at 2000g for 2 min. DNA content analysis was performed using NovoCyte flow cytometer (ACEA Biosciences, San Diego, CA, USA) equipped with the NovoExpress software. The gating strategy for the cell cycle analysis of DNA distribution by flow cytometry is presented in Supplementary Fig. 12.

**Scheme of treatments.** In experiments involving cells synchronized in G0, adenoviruses, siRNA, plasmids, or drugs were applied 36 h before the end of 56 h culture in medium containing 0.1% FCS.

**Adenoviruses.** Cells or human islets were transduced with rodent IAPP (rIAPP) or human IAPP (hIAPP) adenoviruses[8] (75 or 100 MOI [multiplicity of infection] for cells or islets, for 30–36 and 48 h, respectively. The adenovirus-based short hairpin RNA (shRNA) expression system (Ad-RFP-U6-h-HIF1α-shRNA), (Ad-RFP-U6-r-HIF1α-shRNA), (Ad-GFP-U6-r-PFKFB3-shRNA) against human HIF1α, rodent HIF1α and PFKFB3 and control adenovirus (Ad-U6-shRNA-RFP) were purchased from Vectorbiolabs.

**Small interfering RNA.** PFKFB3 small interfering RNAs (siRNAs) (L-095107-02-0005) were purchased from Dharmacon, Lafayette, CO, USA.

**Plasmids.** Drp1 K48A plasmid containing a dominant negative mutation in Drp1 gene was kindly provided by Dr. Takehiro Yasukawa (University College London, London, UK).

**Drugs.** Oligomycin (5 mM) (Sigma 04876, St. Louis, MO, USA) and 2-deoxy-glucose (2-DOG, 1 mM) (Sigma D6134, St. Louis, MO, USA) were used in experiments evaluating the mitochondrial membrane potential. Final concentration of DMSO in medium was <0.04.

**Mitochondrial membrane potential**. Cells synchronized in G1/S or S phase of cell cycle were washed with PBS and trypsinized. One million cells from each sample were incubated for 15 min at 37 °C with TMRE (10 nM, Sigma 87917, St. Louis, MO, USA). Afterwards cells were centrifuged at 2000g for 2 min, TMRE solution was removed and cells were resuspended in fresh culture medium. Mitochondrial membrane potential was measured using NovoCyte flow cytometer (ACEA Biosciences, San Diego, CA, USA). Data were analyzed by NovoExpress software.

**Mitochondrial network**. INS 832/13 cells were grown on coverslips and incubated with the cell-permeant mitochondria-specific red fluorescent probe MitoTracker Red CMXRos (MTR) (Cell Signaling Technology 9082P, Danvers, MA, USA,) at a final concentration of 50 nM at 37 °C for the last 30 min in culture. Cells were then washed with PBS and fixed in 100% methanol at −20 °C for 20 min. Images were taken under a 63× objective with the AxioImager.M2a fluorescence microscope (Zeiss, Oberkochen, Germany) equipped with the optical sectioning system ApoTome.2 and software ZEN2. At least 500 cells per group were analyzed to quantify the mitochondrial architecture. Mitochondrial morphology was classified as fused-to-intermediate if fused mitochondria occupied >50% of the mitochondrial area and fragmented if fragmented mitochondria were present in >50% of the mitochondrial area. Mitochondrial morphology was independently scored by two observers (C.M. and K.V.).

**Calcium measurements**. To measure the concentration of cytosolic-free $Ca^{2+}$, cells or islets were loaded with 2.5 μM Fura 2-AM for 30–45 min, followed by a wash for 10 min at 37 °C. For all measurements in INS832/13 cells, $7 \times 10^5$ cells were seeded onto poly-L-lysine coated coverslips in a 6-well plate 24 h prior treatments. Cells that reached ~60% confluence the next day were either infected with the genetically encoded FRET probe D4ER adenovirus (for measuring ER calcium)[37] or transfected with the ratiometric mitochondrial pericam (Mito-Pericam), for mitochondrial calcium[72] using Lipofectamine 3000 (Thermo Scientific, USA), 2 h prior RNAi silencing with 75 nM of either non-targeting control (CTRL) or rat PFKFB3 siRNA (Dharmacon, USA). Following 8 h of incubation, the cells were then transduced with either LacZ or hIAPP expressing adenovirus (75 MOI each) for 40–43 h. Cells were imaged after incubation for 30 min in a buffer containing 2.8 mM glucose and 0.1% BSA, followed by buffer containing 16.8 mM glucose during imaging. Experiments were carried out at 33.5 °C using an in-line solution and chamber heaters (Warner Instruments, Hamden, CT, USA). Excitation was provided by a TILL Polychrome V monochromator (FEI, Munich, Germany). Excitation (x) or emission (m) filters (Chroma Technology, Bellows Falls, VT, USA) were as follows: Fura-2, 340/10x and 380/10x , 535/30m (R340x/380x –535m); D4ER, 430/24x , 470/24m and 535/30m (430x –R535m/470m); Mito-Pericam 485/15x , 400/15x , t510lpxrxt dichroic, and et535/50m emission filter (Chroma Tech. Corp., USA). Fluorescence emission was collected with a QuantEM:512SC cooled CCD camera (PhotoMetrics, Tucson, AZ, USA) at 10 s interval. Data were acquired and analyzed using Metafluor software (Molecular Devices, Sunnyvale, CA, USA) and plotted using Igor Pro (WaveMetrics Inc., USA).

**Immunocytochemistry and morphometric analysis of cell culture**. 300,000 cells were seeded on coverslips in 6-well plate and synchronized as previously described. Cells were fixed with 4% PFA for 10 min at room temperature. After washing, cells were permeabilized with 0.4% Triton X-100/Tris-buffered saline for 15 min at room temperature, blocked with 3% bovine serum albumin, 0.2% Triton X-100/Tris-buffered saline for 1 h at room temperature and incubated overnight at 4 °C with primary antibodies. Secondary antibodies were applied for 1 h at room temperature. For the PI staining, cells were incubated with 0.5 μM propidium iodide (PI, Molecular Probes, Eugene, OR, USA) for 20 min at 37 °C as previously described[8] and then fixed with 4% PFA. Coverslips or slides were mounted using Vectashield with DAPI (Vector Laboratories, H-1200, Burlingame, CA, USA). The frequency of cell death was evaluated after staining with PI or with MTR. 25 fields per section were imaged using a Leica DM6000 fluorescent microscope (Wetzlar, Germany) with a 20× objective equipped with OpenLab 5.5 software (Improvision, Coventry, UK). Only cells that had two-thirds or more of the nuclear area covered by PI were considered positive. The frequency was expressed as percentage of cells expressing the marker of interest over the total cells counted. Image analysis was performed blindly by two independent investigators (K.V. and C.M.). TUNEL assay was performed with a commercial kit and according to manufacturer's instructions (Roche).

**U-$^{13}$C-glucose tracing and HPLC-MS analysis**. For metabolomic analysis, cells were incubated in medium containing $[U-^{13}C_6]$ glucose (Cambridge Isotope Laboratories CML1396, Tewksbury, MA, USA) for 24 h. To extract intracellular metabolites, cells grown in 6-well plate were briefly rinsed with 2 ml of ice-cold 150 mM ammonium acetate (pH = 7.3), before addition of 1 ml of ice-cold 80% methanol. Cells were scraped and transferred into Eppendorf tubes and then 5 nM D/L-norvaline was added. After vortexing at maximum velocity, samples were spun at 20,000g for 5 min at 4 °C. Supernatant was then moved into a glass vial, dried using speedvac centrifuge, and reconstituted in 50 μl 70% acetonitrile. 5 μl of each sample was injected onto a Luna NH2 (150 mm × 2 mm, Phenomenex, Torrance, CA, USA) column. Samples were analyzed with an UltiMate 3000RSLC (Thermo

Scientific, Waltham, MA, USA) coupled to a Q Exactive mass spectrometer (Thermo Scientific, Waltham, MA, USA). The Q Exactive was run with polarity switching (+3.00 kV/−2.25 kV) in full scan mode with an $m/z$ range of 70–1050. Separation was achieved using 5 mM NH4AcO (pH 9.9) and ACN. The gradient started with 15% NH4AcO and reached 90% over 18 min, followed by an isocratic step for 9 min and reversal to the initial 15% NH4AcO for 7 min.

**Luciferase assay**. A luciferase reporter construct containing PFKFB3 promoter with hypoxia responsive elements (HRE), a luciferase reporter with actin promoter, and an empty luciferase reporter construct were purchased from SwitchGear Genomics (Menlo Park, CA; #S722433, #S717678, and #S790005, respectively).

INS 832/13 cells were transfected with appropriate luciferase plasmids and transduced with LacZ or hIAPP expressing adenoviruses for 36 h. Cells were harvested and treated with LightSwitch Luciferase Assay Reagents (SwitchGear Genomics) according to the manufacturer's instructions. Luciferase signal was measured using SpectraMax L.

**Animal models**. Animal studies were performed in compliance with the guidelines of the UCLA Office of Animal Research Oversight. Ethical approval was obtained from the Research Safety & Animal Welfare Administration at UCLA (ARC #2004-114-51). Wild type (WT) and hIAPP overexpressing (HIP) transgenic male rats between 2 and 6 months of age were generated as previously described[73,74]. Littermates of the same sex were randomly assigned to experimental groups.

**Islet isolation**. After an overnight fast, animals were euthanized using isoflurane. The bile duct was cannulated, and a Hanks' balanced salt solution (HBSS) (Invitrogen, Carlsbad, CA, USA) containing 0.23 mg/ml liberase (Roche 05401020001, Basel, SUI), and 0.1 mg/ml DNase (Roche 10104159001, Basel, Switzerland) was injected in the pancreas. The pancreas was then removed and transferred into a glass vial containing ice-cold liberase solution, digested for 20 min at 37 °C, and dispersed by shaking for 30 s. Islets were manually picked and cultured in RPMI 1640 medium (11 mM glucose) supplemented with 100 IU/ml penicillin, 100 mg/ml streptomycin, and 10% FCS. Islets were studied within 2 days of isolation.

**Lactate measurements**. Medium from cultured rodent islets was sampled every hour within 4 h and lactate was analyzed using an enzymatic assay (Trinity Biotech 732-10, Bray, Ireland) according to the manufacturer's instructions. Islets were collected for protein extraction. Lactate production (per μg protein) was expressed as the hourly change in the accumulated amount of lactate.

**Mitochondrial function**. OCR was determined using the Seahorse XF Extracellular Flux Analyzer (Seahorse Bioscience, North Billerica, MA, USA). After an overnight recovery, isolated islets from WT and HIP rats were seeded (25–50 islets per well) into the V7 plate (Seahorse Bioscience, North Billerica, MA, USA). To assess mitochondrial function, OCR was measured at the basal state and after stimulation with 20 mM glucose and sequential injection of oligomycin (ATP synthase inhibitor), carbonyl cyanide-p trifluoromethoxyphenylhydrazone (FCCP; uncoupler), and rotenone (complex I inhibitor).

**Human subjects**. Pancreata from brain dead (cadaveric) organ donors with diabetes for whom consent has been provided from the relatives by the donors were obtained from the Network for Pancreatic Organ Donors with Diabetes (nPOD), administered by the University of Florida, Gainesville, Florida, in a collaborative manner. All procedures were in accordance with federal guidelines for organ donation and the University of Florida Institutional Review Board. Three pancreata from individuals with T2D (6186, 6275, 6255) and 3 from ND (6104, 6288, 6020) controls matched by age, sex, and BMI were examined in this study.

**Human islets**. Human pancreatic islets were from the Islet Cell Resource Consortium. Using of human islets was determined as not human subjects research by the UCLA Office of Human Research Protection Program (IRB Exception, 01.06.2016). They were derived from brain-dead (cadaveric) organ donors, for whom consent has been provided by the relatives, through the Integrated Islet Distribution Program (IIDP) in a collaborative manner. IIDP is supported by the National Institute of Diabetes and Digestive and Kidney Diseases (NIDDK) and provides a network for pancreatic islet availability for fundamental research. Dithizone staining was performed to assess the islet purity that was 90–95%. The donors, aged 25–60 years, were heart-beating cadaveric organ donors. Islets were cultured in RPMI 1640 medium (5.5 mM glucose) containing 100 units/ml penicillin, 100 g/ml streptomycin, and 10% fetal bovine serum (Invitrogen, Carlsbad, CA, USA) for 1 day and then processed for western blotting analysis.

**Antibodies**. For detection of PFKFB3, we utilized a previously reported antibody[75], anti-PFKFB3 (Abcam 181861, Cambridge, UK, 1:200 for IF, 1:1000 for WB) the specificity of which we confirmed by silencing PFKFB3 and measuring transcript and protein levels by western blot, qRT-PCR, and immunofluorescence (Supplementary Fig. 11a-d).

The following antibodies were used: anti-PFKFB3 (Abcam 181861, Cambridge, UK, 1:400 for IHC, 1:200 for IF, 1:1000 for WB), anti-Tom20 (Santa Cruz Biotechnology sc-11415, Dallas, TX, USA, 1:200 for IF), anti-HIF1α (NOVUS Biologicals NB100-105, Littleton, CO, USA, 1:1000 for WB), anti-MFN2 (Cell Signaling Technology 9482S, Danvers, MA, USA, 1:1000 for WB), anti-Opa1 (BD Transduction 612606, San Diego, CA, USA, 1:1000 for WB), anti-Drp1 (Cell Signaling Technology 8570S, Danvers, MA, USA, 1:1000 for WB), anti-nucleolin (Santa Cruz Biotechnology sc-13057, Dallas, TX, USA, 1:1000 for WB), anti-PARP1 (Cell Signaling Technology 9542S, Danvers, MA, USA, 1:1000 for WB), anti-insulin (DAKO A0564, Glostrup, Denmark, 1:200 for IF), anti-GAPDH and anti-cleaved caspase 3 from Cell Signaling Technology 2118S and 9664S, respectively, Danvers, MA, USA, (1:1000 for WB). Secondary antibodies for immunofluorescence staining were F(ab')2 conjugates with Cy3 or FITC purchased from Jackson Laboratories and used at dilution of 1:200.

Uncropped and unprocessed Western blots are provided in the separate Data Source File.

**qRT-PCR.** The levels of PFKFB3, LDHA, and MCT1 mRNA were quantified by qRT-PCR. Total RNA was isolated using a RNeasy mini kit (Qiagen, Hilden, Germany) according to the manufacturer's instructions. 250 ng of total RNA from each sample was denatured at 65 °C and then reverse transcribed using Superscript III reverse transcriptase (Invitrogen, Carlsbad, CA, USA) at 50 °C for 1 h. Real-time quantitative polymerase chain reaction (qPCR) was performed using ABI7900HT (Applied Biosystems™, Foster City, CA, USA) with initial denaturation at 95 °C for 20 s, followed by 45 cycles of 94 °C for 1 s and 60 °C for 20 s, then continued with a dissociation stage. Each qPCR reaction contained 1× Fast SYBR® Green Master Mix (Applied Biosystems™, Foster City, CA, USA), 1 μM of each primer, and 400 ng cDNA. Relative mRNA expression of target gene was determined using the comparative cycle threshold (Ct) method, where the amount of target cDNA was normalized to the internal control, GAPDH cDNA. The primers used were: PFKFB3 (fwd: CACGGCGAGAATGAGTACAA, rev: TTCAGCTGACTGGTCCA CAC)[76]; LDHA (fwd: TGC TGG AGC CAC TGT CG, rev: CTG GGT TTG AGA CGA TGA GC)[77]; MCT1 (fwd: ATG TAT GCC GGA GGT CCT ATC, rev: CCA ATG GTC GCT TCT TGT AGA)[78]; and GAPDH (fwd: ATG ACT CTA CCC ACG GCA AG, rev: CTG GAA GAT GGT GAT GGG TT).

**Tissue immunostaining and morphometrical analysis.** 4-μm paraffin tissue sections from human or rodent samples were exposed to toluene for 10 min and, then, to 100% ethanol for other 10 min, 95% ethanol and 70% ethanol for 5 min each, and water. Sections were transferred in heat-induced antigen retrieval solution in citrate buffer at pH 6.0, using microwave and then cooled to room temperature for 1 h, then soaked in Soaking Buffer (TBS, 0.4% TX100) for 30 min on ice, and washed once with TBS. After blocking the unspecified binding sites with a blocking solution (TBS, 3% BSA, 0.2% TX100) for 1 h, the slides were incubated with the primary antibodies diluted in Antibody Buffer (TBS, 3% BSA, 0.2% Tween-20) overnight at +4 °C. After washing in TBST, slides were incubated with secondary antibodies diluted in Antibody Buffer for 1 h at room temperature. Slides were then mounted using Vectashield with DAPI. The presence of PFKFB3 in the islets was evaluated in pancreatic sections immunostained for PFKFB3 and insulin. Images of 25 islets per sample were taken using a Leica DM6000 fluorescent microscope (Wetzlar, Germany) with a 20× objective equipped with OpenLab 5.5 software (Improvision, Coventry, UK). The frequency of nuclear PFKFB3 staining was expressed as a percentage of β-cells expressing PFKFB3 only in the nuclei. Nuclei were considered positive for PFKFB3 staining only if at least two-thirds of their area was occupied by multiple bright puncta of the marker of interest. The frequency of both cytoplasmic and nuclear PFKFB3 staining was expressed as percentage of β-cells expressing the PFKFB3 in the cytoplasm and nuclei. Image analysis was performed blindly by two independent investigators (K. V. and C.M.). To visualize mitochondria in human pancreatic tissue, sections were stained with Tom20. The mitochondrial area was quantified using the Image-Pro Premier 9.1 software (Rockville, MD, USA) and expressed as Tom20 positive area, inside the insulin positive area in the islet, divided by the number of β-cells.

**Western blotting.** To prepare whole cell extracts, cells or islets were incubated for 20 min on ice in NP40 lysis buffer (20 mM Tris–HCl, 150 mM NaCl, 2 mM MgCl$_2$, 0.5% NP-40, 1 mM DTT, 5 mM NaF, 1 mM Na$_3$VO$_4$), and protease inhibitor cocktail (Sigma P2714, St. Louis, MO, USA), sonicated, and spun at 10,000g at 4 °C for 10 min. To separate cytoplasmic and nuclear protein fractions, after incubation in NP40 lysis buffer, samples were centrifuged at 3500g at 4 °C for 10 min. Then, supernatant representing the cytoplasmic part was transferred in another Eppendorf whereas the pellet (nuclear part) was resuspended in RIPA lysis buffer. Protein concentration was determined using the DC protein assay kit (Bio-Rad, Irvine, CA, USA). Proteins (20–35 μg/lane) were separated by SDS-PAGE (4–20%) and, then, transferred onto polyvinylidene fluoride membranes (Bio-Rad, Irvine, CA, USA) by semi-dry electroblotting. After blocking with 5% milk for 1 h, membranes were probed overnight at 4 °C with primary antibodies. Then, membranes with transferred protein were incubated with horseradish peroxidase-conjugated secondary antibodies for 1 h at room temperature (Invitrogen, Carlsbad, CA, USA). Proteins were visualized using ECL reagents from Bio-Rad and expression levels were quantified using Labworks software (UVP).

**Statistical analysis.** Results are expressed as the means ± SEM. The statistical analysis was performed by two-tailed $t$ test, one-way ANOVA, or two-way ANOVA with repeated measures using GraphPad Prism software (La Jolla, CA, USA). A value of $p < 0.05$ was taken as an evidence of statistical significance.

**Reporting summary.** Further information on research design is available in the Nature Research Reporting Summary linked to this article.

## Data availability
The data that support the findings of this study are available from the corresponding author upon reasonable request. Source data for Figs. 1, 2, 3, and 9 are provided in the separate Data Source File. The metabolomics data set was deposited in the MetaboLights under identifier MTBLS951. Study dataset can be accessed at https://www.ebi.ac.uk/metabolights/MTBLS951.

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

## Acknowledgements

This work was supported by funding from the National Institutes of Health (NIH/NIDDK Grant #DK077967) and the Larry Hillblom Foundation (Grant #2014-D-001-NET and Start-up Grant #2017-D-002-SUP). The work in Dr. Satin's lab was supported by NIH/NIDDK Grant #DK46409. V.S.P. was supported by an Upjohn Postdoctoral Fellowship. L.P. was supported by Department of Endocrinology, Shengjing Hospital of China Medical University, Shenyang, China. This research was performed with the support of the nPOD, a collaborative T1D research project sponsored by the Juvenile Diabetes Research Foundation International. Organ Procurement Organizations (OPO) partnering with nPOD to provide research resources are listed at www.jdrfnpod.org/our-partners.php. The authors thank Dr. Arthur Sherman for discussions of cellular $Ca^{2+}$ dynamics and statistics, Benjamin Thompson for his help with the experiments involving $Ca^{2+}$ measurements, and UCLA Metabolomics Center for generation of metabolomics data. Dr. Butler's grant was funded by the National Institute of Diabetes and Digestive and Kidney Diseases, #DK059579.

## Author contributions

C.M., V.S.P., K.E.V., L.P., and H.N. performed the experiments and contributed to data analysis. V.S.P. and S.V. conducted the calcium experiments. E.R. contributed to perform the mitochondrial respiration assay. A.E.B., T.G., and O.S.S. contributed to the experimental design, results, and manuscript writing. V.S.P. and L.S.S. contributed to experimental design and interpretation of the calcium data and to the writing of the manuscript. P.C.B. and S.T. gave the initial idea for the project, supervised the experimental work, and wrote the manuscript.

## Additional information

**Competing interests:** The authors declare no competing interests.

