## [Peer Review File · Nature Communications]

Reviewer #1 (expert in pancreatic beta cell function and proliferation)

(Remarks to the Author):

Re: Targeting PFKFB3 rescues beta-cells from islet amyloid pancreatic polypeptide (IAPP) toxicity

In this manuscript, Montemurro et al. suggested beta cell dysfunction in T2D is provoked by mitochondrial fragmentation resulting from IAPP toxicity. Overexpressing hIAPP in rats and INS1 cells activated the HIF1a/PFKFB3 stress pathway, leading to abnormal cytosolic Ca²⁺ homeostasis, loss of insulin secretion and beta-cell death. Furthermore, inhibiting/ silencing the expression of PFKFB3 in INS1 cells that have been transduced with hIAPP adenovirus can rescue the mitochondrial network fragmentation and reduce beta cell death phenotypes associated with ectopic expression of hIAPP, marking it as a potential therapeutic target in protecting mitochondrial integrity. The authors argue that owing to the commonalities in mitochondria structural and functional deterioration in diabetic beta cells and neurodegenerative diseases, uncovering this mechanism will provide important insight in both settings.

However, the manuscript is sprawling and difficult to follow, it also falls short in rationale and system choices. The human protein should be studied in human islets. There is insufficient evidence to support the proposed mechanism in IAPP-mediated cell death in rat INS1 cells.

Major points

1. The authors need to indicate the number of donors samples used for staining in Fig 1a.
2. MTR is not an appropriate stain used in Fig. 1d-e. It is a potentiometric dye and it marks areas of the network that maintain potential across the network. The authors need to repeat and confirm this with the TOMM20 antibody. In addition, the novelty/significance of the fragmentation data is questionable: it is well established in the field for years that mitochondrial fragmentation is an early event in apoptosis.
3. The authors used islets isolated from transgenic rat expressing human IAPP driven by RIPII promoter (RIP-II h-IAPP) (Fig 3D) and hIAPP overexpressing INS1 cell (Fig S4A) and showed increased PFKFB3 protein levels, then change systems and report increased PFKFB3 and PFK1 protein levels islets isolated from transgenic mice expressing Ins2-IAPP but data not shown (line 239 – 241). What is the reason for this system change? The PFK1 data in the transgenic rat should also be included in Fig 3.
4. The data and conclusion given in Fig. 4 are not convincing. The authors should provide evidence for the specificity of HIF1a binding and regulation of PFKFB3, in both mRNA and protein levels:

- a. The authors show enhanced PFKFB3 staining in transgenic rat (Fig4a) and T2D (Fig4b) beta cells. The authors need to demonstrate if this increase expression is dependent on HIF1a by silencing HIF1a in rat and T2D islets followed by staining and quantifying PFKFB3 positive cells.
 - b. HIF1a binds to PFKFB3 promoter and promotes PFKFB3 expression, yet increasing HIF1a protein levels does not necessarily increase PFKFB3 protein expression. To show HIF1a specificity and to establish a mechanistic connection as suggested in line 247, the authors need to do CHIP to show that HIF1a binds to PFKFB3 HREs promoter in beta cell; they also need to silence HIF1a in both rat (WT and HIP) and human islets (ND and T2D), perform qRT-PCR and WB to observe the changes in both PFKFB3 mRNA and protein levels.
5. The authors showed evidence HIAPP toxicity caused mitochondrial fragmentation and abnormal cytosolic calcium levels, and these phenotypes can be rescued through silencing PFKFB3. The authors suggest that silencing PFKFB3 should also improve insulin secretion, but they do not show these data. They need to perform GSIS in WT, HIAPP and HIAPP+PFKFB3 siRNA treated INS1 cells to make this conclusion.
6. Quantitation for the Western Blots as shown in Fig. S2A (and throughout the manuscript) needs to be repeated. The authors should perform the assay for at least three independent trials. GAPDH is not an appropriate control for western blots where metabolism changes are also occurring as it is metabolically regulated.
7. Potential across the IMM is required for ongoing fusion. It is more likely that loss of potential in regions of the network (where MTR staining is lost) is the reason for the loss of fusion, not alterations in levels of MFN proteins.

Minor points

1. The authors adapted partial results from Schludi et al. in Fig. 3a, credit should be given in both text and figure legend (line 1071).
2. The authors need to explain the doublet vs singlet bands seen in Cl. Casp-3 in Fig. 6C.
3. Figures S3e and 7a are repetitive; there is no need to present the same schematic figure twice.
4. The reason for the nomenclature change from HIAPP to HIP is unclear. The authors should also provide proper nomenclature of animal models used in their study.
5. Change to number reference in line 329.
6. Typo in line 663, donor not donors.
7. It should be Fig. 2a instead of Fig. 1a in line 1062.
8. Unreadable characters in lines 493, 677, 679, 739

Reviewer #2 (expert in IAPP)(Remarks to the Author):

The authors employ gene array and metabolomics approaches to identify potential regulators of human IAPP-induced toxicity in pancreatic beta cells. The findings link toxicity of human IAPP aggregates to mitochondrial disruption and glycolysis, in particular expression of PFKFB3. Of note, some of the histological findings are also found in pancreatic islets from type 2 diabetes. Although such findings are of some interest and novelty, the data as presented do not fully support the conclusions. A number of overstatements and broad conclusions are made, requiring 'connecting of the dots' such that their model diagram (Fig. 7) needs to be much better supported by data. The findings of changes in islet cell metabolism described in HIP rat islets may be secondary to hyperglycemia following loss of beta cell mass due to increased cell death from IAPP induced toxicity.

Specific criticisms

1. Important conclusions are made based on changes in PFKFB3 expression that are not fully supported by the data provided:

(a) Timing of changes in PFKFB3 expression: the authors state that increased islet expression of PFKFB3 precedes onset of hyperglycemia at (6 months) in HIP rats. This could be important if true, but the data do not clearly show this. First, PFKFB3 mRNA levels at 6 months (when hyperglycemia is said to be present) are clearly elevated, but this is not obviously the case at 1 or 3 months (Fig 3D), where at each time point only one of two (only N=2) HIP bands appear denser. In a previous publication, blood glucose levels were elevated at 5 months, prior to any clear increase in PFKFB3 expression (based on the data provided). Importantly, blood glucose data from the current studies are not provided for comparison. Second, the PFKFB3 protein data are described as 2-fold greater prior to hyperglycemia but reported as 'not shown'. When there is supplementary data and therefore no reason to not show data, and in keeping with Nature Comm journal policy to avoid 'data not shown', and given that these data are important to the overall message of the importance of PFKFB3, these data should be provided and accurately described.

(b) Localization of PFKFB3 expression: By immunostaining, PFKFB3 immunoreactivity appears to be largely nuclear in HIP and T2D islets. The authors ascribe the changes in glycolysis in large part to the increase in PFKFB3 expression. Nuclear PFKFB3 has been described by others as a mechanism to increase cell proliferation in response to glucose. Since glycolysis is presumed to occur in the cytoplasm, and how do the authors reconcile increased nuclear (but possibly not cytoplasmic) expression of PFKFB3 with increased glycolysis? In the model in Fig 7, the PFKFB3 is in the cytoplasm, not the nucleus, which seems an inaccurate depiction of the data. Is nuclear translocation of PFKFB3 occurring?

(c) Silencing of PFKFB3 expression: The authors show that silencing PFKFB3 restores mitochondrial fragmentation (Fig 6), beta cell survival and gene expression in INS-1 cells, but show no effects on glycolysis, cell respiration/OCR, lactate or ATP production or anything in primary islets. Such data are critical to the overall conclusions of the paper. It is also important to show from a therapeutic perspective that silencing PFKFB3 does not have deleterious effects (or perhaps improves) beta cell function (glucose-stimulated insulin secretion).

2. The connection of HIF and γ H2A.X to the observed changes are tenuous. γ H2A.X is said to be elevated in pre-diabetic HIP rats, although the data provided in Fig 3E are at 6 months when these rats are said to already be diabetic (though again, no glucose data are provided). It seems equally likely that the observed changes are secondary to human IAPP induced loss of beta cell mass (as is observed in HIP rats), and hyperglycemic stress on remaining beta cells. The authors state: "...these findings indicate that hIAPP decouples glycolysis from oxidative respiration due to γ H2A.X-associated accumulation of nuclear HIF1 α in INS 832/13 cells and β -cells from hIAPP transgenic rodents" and "We also establish that ...this metabolic adaptation is mediated at least in part by the activation of the HIF1 α /PFKFB3 stress pathway" yet the role of HIF1 α in the observed changes is not shown, rather just correlative. These statements need to be softened or data provided; for example, the temporal relationship of HIF1 α , PFKFB3, and γ H2A.x could be addressed in Fig 3D showing changes in HIF1 α and γ H2A.x protein over time.

3. While the authors conclude that hIAPP oligomers mediate the changes observed, this was not directly demonstrated, but rather appears to be based on their previous data showing oligomer formation in these models. They should either show oligomers histologically in cells associated with the metabolic changes and mitochondrial disruption, or more precisely state that the observations are associated with "human IAPP overexpression", or "human IAPP toxicity" or perhaps IAPP "misfolding" or "aggregates", since the species is not identified in these experiments. As examples:

(a) Results section subtitle: "hIAPP toxic oligomers induce mitochondrial network fragmentation with reduced mitochondrial function."

(b) Discussion: "we uncovered that stress induced by hIAPP toxic oligomers recapitulates the metabolic phenotype reported in β -cells in T2D"

4. The array data and other studies are performed on whole islets, with conclusions that findings are occurring in beta cells. While the conclusion that these changes are occurring in beta cells is supported by complementary studies in INS-1 cells overexpressing human IAPP by adenovirus, it remains possible that changes in gene expression are occurring in non-beta cells. This should be clearly stated.

5. Given the evidence (and conclusion) of a marked switch to aerobic glycolysis and production of lactate associated with human IAPP overexpression, and use of Seahorse to assess oxygen consumption, it would seem an easy, complementary, and useful measure for the authors to provide data of extracellular acidification rate (ECAR) from HIP and wild-type rat islets. Was acidification changed along with OCR and lactate production?

6. Fig 6D: human IAPP expression decreases cytosolic calcium and PFKFB3 increases it. To this reviewer, this does not make any sense in terms of the conclusions and the model in Fig. 7.

7. The TMRE data are also difficult to interpret as presented. First, there is no direct comparison between human IAPP and control cells, although it is said (data now shown) to be not different. Second, there is no shift in TMRE intensity, only a decrease in cell count. This suggests no difference in mitochondrial membrane potential (the driving force for ATP production) between human IAPP overexpressing and control beta cells, which seems surprising considering the conclusion that human IAPP overexpression is driving glycolysis, ATP production, and closure of potassium-sensitive ATP channels. One possible interpretation is that human IAPP overexpressing cells are more sensitive to cell death induced by any toxic stimulus (including DOG and oligomycin). Do the human IAPP expressing cells have increased sensitivity to any toxic stimulus? The TMRE data need to be interpreted in terms of their meaning to the proposed model of glycolysis and ATP production, since as presented the data do not seem to support this model. Ideally, ATP would be measured.

Minor comments:

1. Fig S1 should clearly state what the control is, presumably rat IAPP adenovirus
2. Dotted lines should be used in Fig 7 for those aspects of the model that were not shown by the data in the current manuscript.
3. Why does PFKFB3 silencing increase G2/M – is it increasing cells in DNA damage checkpoint? Or increasing survival?

Reviewer #3 (expert in intracellular metabolism of beta cells)(Remarks to the Author):

The manuscript by Montemurro et al. assesses the mechanisms responsible for the toxic effects of human IAPP in β -cells. The authors perform studies mainly in insulinoma cells complemented with some studies in a rat model overexpressing hIAPP. Some studies using human islets are also included. The authors conclude that hIAPP induces major metabolic and mitochondrial network changes through the activation of the HIF1 α /PFKFB3 stress pathway. The authors propose that this pathway in the short term is aimed to preserve β -cell survival but in the long term result in cell death mediated by cytosolic Ca²⁺ accumulation.

1. The experimental designed is well thought out and the experiments are well designed. However, the findings in the study are for the most part confirmatory of data in the neuronal literature where similar processes occur in neurodegenerative disorders.

2. In addition, many laboratories have shown that hIAPP induce beta cell dysfunction and death by inducing several mechanisms including oxidative stress, autophagy, mitochondrial dysfunction and ER stresses. How the current findings fit into the previous mechanisms is unclear.

3. The authors use three models in these studies including INS cells, islets from HIP rats and human islets. The resilience in the use of INS1 cells without validation in the transgenic rat is concerning. There is no data on how the hIAPP is introduced to the cells and the cited reference does not include the description of the cells. This is important, as there are limitations of transformed cells INS1 (tumoral cells) that could potentially alter intracellular metabolism. In addition, it is unclear if the degree of hIAPP overexpression between the INS1 model and the HIP islets is similar. No immunoblotting comparing levels of hIAPP was presented. Did the authors replicate findings by treatment with hIAPP?

4. The authors focused on studying the effects of PFKFB3 as a mechanism for the phenotype. Interestingly, no quantification from the immunoblotting is presented and only two biological variables are included. Similarly, immunoblotting for PFKFB3 from human islets is not particularly convincing. Does hIAPP overexpression (as shown by the authors previously) in human islets reproduce this effect?? Given the numerous published mechanisms implicated in hIAPP induce beta cell dysfunction and apoptosis, it is unclear if this mechanism is a major component for regulation of beta cell mass in vivo. Rescue experiments by hIAPP transgenic mice to PFKFB3 heterozygous mice (16715124). Without this evidence it is unclear to determine the role of this process in vivo protection from diabetes induced by hIAPP.

5. Mitochondrial function can be regulated by many mechanisms. The authors show mitochondrial dysfunction at 4-6 months of age (4 or 6m?). Do this occur earlier? Why does it takes 4-5 months to develop if hIAPP is expressed from development? Therefore, the authors should perform mitochondrial function studies at earlier stages to provide stronger evidence for this mechanism in the phenotype. The authors study fusion/fission in INS1 cells and function in islets. This makes the conclusion less convincing and data on fusion/fission should be include from dispersed beta cells from HIP rats.

6. The metabolic studies are exclusively performed in INS1 cells. Are these defects reproduced in islets beyond lactate levels. Key metabolic flux findings should be replicated in islets from HIP model. Similarly, calcium measurements should also be performed in islets from HIP rats.

Minor

1. Reference for INS1 cells overexpressing hIAPP is wrong

2. Table with more information on islet donors is missing. Diabetes duration, etc??

Reviewers' comments:

Reviewer #1 (expert in pancreatic beta cell function and proliferation) (Remarks to the Author):

Re: Targeting PFKFB3 rescues beta-cells from islet amyloid pancreatic polypeptide (IAPP) toxicity

In this manuscript, Montemurro et al. suggested beta cell dysfunction in T2D is provoked by mitochondrial fragmentation resulting from IAPP toxicity. Overexpressing hIAPP in rats and INS1 cells activated the HIF1 α /PFKFB3 stress pathway, leading to abnormal cytosolic Ca²⁺ homeostasis, loss of insulin secretion and beta-cell death. Furthermore, inhibiting/silencing the expression of PFKFB3 in INS1 cells that have been transduced with hIAPP adenovirus can rescue the mitochondrial network fragmentation and reduce beta cell death phenotypes associated with ectopic expression of hIAPP, marking it as a potential therapeutic target in protecting mitochondrial integrity. The authors argue that owing to the commonalities in mitochondria structural and functional deterioration in diabetic beta cells and neurodegenerative diseases, uncovering this mechanism will provide important insight in both settings.

However, the manuscript is sprawling and difficult to follow, it also falls short in rationale and system choices. The human protein should be studied in human islets. There is insufficient evidence to support the proposed mechanism in IAPP-mediated cell death in rat INS1 cells.

We thank the reviewer for the constructive and helpful feedback. By using this feedback and generating new experimental findings, we feel we have improved the manuscript and developed a more cohesive model to account for the metabolic changes in cells confronted by the misfolding of amyloidogenic proteins. In short, we now propose that the metabolic and mitochondrial network changes that occur in β -cells in type 2 diabetes represent the initial steps of a conserved stress repair response. As such, these are adaptive changes that serve to initially protect cells, from dying in response to the injury. We now postulate that, in common with other cell types having minimal capacity for cell replication (such as neurons), β -cells in type 2 diabetes remain trapped in the initial stages of this stress regeneration program rather than executing the second stage in which cells enter cell cycle. This critical second step of the injury regeneration pathway takes advantage of the dual role of the cell cycle to eliminate cells with DNA damage (that fail cell cycle checkpoints) while regenerating tissue loss with healthy cells that complete cell cycle.

This new insight came about through use of both unbiased metabolomics and transcriptional array (Microarray analysis) investigation that permitted us to establish that many of the apparently disparate findings reported in β -cells in type 2 diabetes are anticipated initial responses to the conserved HIF1 α mediated stress response pathway. In contrast to injured tissue with the capacity to regenerate through cell replication (for example the proximal renal tubule following acute tubular necrosis), β -cells in type 2 diabetes are unable to traverse cell cycle to clear damaged cells and restore tissue with new undamaged cells. On the positive side, the HIF1 α stress response pathway reduces the loss of β -cells explaining the notably slow rate of β -cell loss in type 2 diabetes. Unfortunately, on the negative side, since β -cells rely on tight connection of glycolysis with oxidative metabolism to couple circulating glucose with insulin secretion, β -cells trapped in the first phase of the injury response pathway are dysfunctional, explaining impaired glucose mediated insulin secretion in type 2 diabetes.

Several interesting clinical insights emerge from this revised model to explain changes in metabolism in β -cells in type 2 diabetes. First, caution should be exercised implementing strategies that seek to reverse these metabolic changes (for example reversing adaptive mitochondrial form and function) without removing the source of injury (toxic protein oligomers) since this approach may accelerate cell death. Likewise, strategies that seek to drive β -cells through cell cycle without removing toxicity is likely to lead to a loss rather than an increase in cell mass since the injured cells would be expected to fail cell cycle checkpoints and undergo apoptosis rather than replication.

Major points

1. The authors need to indicate the number of donors samples used for staining in Fig 1a.

Thank you. We now provide a table (Table 1) with the requested information. As stated the donors came from the Network Of Pancreas Donors (nPOD). The nPOD identification numbers of donor pancreata as well as available biometric information are in the table , which is shown below for convenience.

Fig. 2B

Fig. 2. HIF1a-PFKFB3 stress pathway is upregulated in β -cells from humans with type 2 diabetes (T2D). (B) Representative immunofluorescence images of islets from non-diabetic (ND) and T2D patients stained for PFKFB3 (red),

Fig. 4B

Fig. 4. β -cell mitochondrial fragmentation in T2D is reproduced in hiAPP model. (B) Representative immunofluorescence images of islets non-diabetic (ND) and T2D patients stained for Tom20 (mitochondria, red), insulin (green) and DAPI (nuclei, blue).

TABLE 1.						CONTROLS (Non-diabetic)			
T2D									
nPOD	Age	Sex	BMI	Duration	Diabetes	nPOD	Age	Sex	BMI
Case No.	[years]			[years]	Medications	Case No.	[years]		
6186	68.4	M	20.9	5	Sitagliptin, Metformin	6104	41	M	20.5
6275	48	M	41	2	None	6288	55	M	37.7
6255	55	M	29.4	6	Metformin Glibenclamide	6020	60	M	29.8
Mean	57.1		30.4	4.3			52		29.3
SEM	5.9		5.82	1.2			5.7		4.9

2. MTR is not an appropriate stain used in Fig. 1d-e. It is a potentiometric dye and it marks areas of the network that maintain potential across the network. The authors need to repeat and confirm this with the TOM20 antibody. In addition, the novelty/significance of the fragmentation data is questionable: it is well established in the field for years that mitochondrial fragmentation is an early event in apoptosis.

The reviewer raises an excellent point regarding loss the use of MTR to evaluate mitochondrial morphology. Therefore, as requested, we also evaluated the mitochondrial network morphology using TOM20 staining. The new data confirms that

the mitochondrial network is indeed fragmented in response to hIAPP overexpression (Fig. 1D and E now replace Fig. 4A).

Fig. 4A-C

Fig. 4. β -cell mitochondrial fragmentation in T2D is reproduced in hIAPP model. (A) Representative images of INS 832/13 cells stained for Tom20 (mitochondria, red) and DAPI (nuclei, blue). Cells were cultured in RPMI medium, synchronized at G1/S of the cell cycle and transduced for 36h with adenoviral vectors expressing LacZ (CTRL) or rodent hIAPP (rIAPP) or hIAPP (hIAPP). (C) Quantification of mitochondrial morphology in G1/S enriched INS 832/13 cells after indicated treatments to overt fragmented or overt intermediate-to-fused mitochondria. Data are presented as mean \pm SEM, n=3 for each group, ***p<0.005, **p<0.01 relative to CTRL, and ##p<0.01 relative to rIAPP.

We also expanded the studies on mitochondrial membrane potential using a TMRE probe as shown in Figure 5C-F. Of interest, hIAPP overexpressing cells have sustained rather than reduced mitochondrial membrane potential as measured by flow cytometry (Fig. 5D). This is consistent with our revised appreciation of the changed mitochondrial network morphology being a component of the adaptive protective changes in response to toxicity rather than an initial driver of toxicity. Investigators in the neurodegeneration field have come to the same conclusion, noting that when mitochondria adopt the fragmented perinuclear form they are more resistant to taking up cytosolic Ca^{2+} and therefore less likely to induce the mitochondrial pathway of apoptosis (Frieden M et al., J Biol Chem, 2004; Lee HC and Wei YH, J Biomed Sci, 2000; Chang DT et al., Prog Neurobiol, 2006; Forte M et al., Novartis Found Symp, 2007; Rintoul GL et al., Biochim Biophys Acta, 2010). Moreover, this pattern of mitochondrial network change is consistent with the observed metabolic switch from oxidative metabolism of glucose to an increased flux through glycolysis that is disengaged from the TCA cycle, but rather diverted to lactate production. Under these conditions, mitochondria preserve membrane potential by use of ATP generated through increased glycolysis.

Fig. 5D

Fig. 5. Mitochondrial respiration but not mitochondrial membrane potential is decreased by hIAPP. (D) Mitochondrial membrane potential was measured by flow cytometry after labelling with TMRE dye in CTRL vs hIAPP overexpressing cells synchronized in G1/S of the cell cycle.

Finally, with regard to the reviewers' good suggestion to assure that the observed fragmented mitochondrial network is not simply a manifestation of apoptosis, mitochondrial fragmentation (and membrane potential) was evaluated in

synchronised cells at G1/S and in both control (CTRL) and *hiAPP* overexpressing cells when apoptosis was below 1%. Therefore, mitochondrial fragmentation as a result of *hiAPP* overexpression was decoupled from apoptosis. We have added these new experiments to the results section (Fig. S3C, and page 7, paragraph 149-151).

Fig. S3C

Fig. S3. *hiAPP* induces fragmentation that precedes cell death in INS 832/13 cells synchronized at G1/S. (C) FlowJo overlay of flow cytometry diagrams from INS 832/13 cells transduced with LacZ (control) or *hiAPP* expressing adenoviruses synchronized at G1/S stage of cell cycle.

3. The authors used islets isolated from transgenic rat expressing human IAPP driven by RIP-II promoter (RIP-II h-IAPP) (Fig 3D) and *hiAPP* overexpressing INS1 cell (Fig S4A) and showed increased PFKFB3 protein levels, then change systems and report increased PFKFB3 and PFK1 protein levels in islets isolated from transgenic mice expressing Ins2-IAPP but data not shown (line 239 – 241). What is the reason for this system change? The PFK1 data in the transgenic rat should also be included in Fig 3.

Thank you for this question. Whenever possible, we typically examine hiAPP induced changes not only using the INS cell model but also in hiAPP transgenic rodents to assure that the changes are also present in primary β-cells as well as in human pancreas to assure potential clinical relevance. For this reason, after observing activation of the HIF1α-PFKFB3 stress pathway in INS 832/13 cells transduced with hiAPP (a prominent signal in unbiased microarray analysis), we corroborated these findings in primary β-cells of hiAPP transgenic rats and humans with type 2 diabetes using western blot and immunohistochemistry techniques.

The hiAPP transgenic mice develop diabetes rapidly (by about 12 weeks of age) while the HIP rats do not develop diabetes until about 10 months of age and have pancreatic islets that resemble human islets (islet amyloid and gradual β-cell loss). The advantage of the first model is obviously cost and rapidity in the execution of experiments while the second model allows us to study the effects of hiAPP oligomers in β-cells during a longer prediabetic condition excluding the confounding effects of the hyperglycemia that can be present in the first model.

As requested we now show increased expression of HIF1α and PFKFB3 in the hiAPP transgenic rats in the revised Figure 1D, E.

Fig. 1D-E

Fig. 1. *hiAPP* leads to upregulation of the HIF1α-PFKFB3 stress pathway and increases aerobic glycolysis. (D) Representative Western blot of PFKFB3 and HIF1a protein levels in whole cell extracts and nuclear enriched fractions of islets from 6 months old WT (3) and HIP (3) rats. GAPDH and PARP were used as loading controls for cytosolic and nuclear extracts, respectively. **(E)** Quantification of HIF1α (upper panel) and PFKFB3 (lower panel) in cytoplasmic and nuclear fractions. Data are presented as mean ± SEM, n=3 for each group, *p<0.05.

4. The data and conclusion given in Fig. 4 are not convincing. The authors should provide evidence for the specificity of HIF1 α binding and regulation of PFKFB3, in both mRNA and protein levels:

a. The authors show enhanced PFKFB3 staining in transgenic rat (Fig4a) and T2D (Fig4b) beta cells. The authors need to demonstrate if this increase expression is dependent on HIF1 α by silencing HIF1 α in rat and T2D islets followed by staining and quantifying PFKFB3 positive cells.

We regret that the reviewer did not find the data in the previous version to not be convincing. In response, we have now evaluated PFKFB3 expression after silencing HIF1 α in non diabetic human islets transduced with hIAPP expressing adenovirus. These new data (Fig. 3B) show that HIF1 α silencing in INS 832/13 cells reduces PFKFB3 expression and LDHA levels and as presented in Supplementary Data Figs. 2A-B show that HIF1 α silencing in human islets overexpressing hIAPP reduces the PFKFB3 expression in β -cells by immunohistochemistry and WB.

Fig. 3B

Fig. 3. HIF1 α drives the expression of PFKFB3 in β -cells. (B) Representative Western blot of HIF1 α , PFKFB3, LDHA, Cdh1 in whole cell extracts of INS 832/13 overexpressing LacZ (CTRL) or hIAPP, silenced or not with short hairpin RNA for PFKFB3 (PFKFB3 shRNA) and HIF1 α (HIF1 α shRNA) for 36h. CTRL shRNA is non target shRNA control vector. B-actin and GAPDH were used as loading control.

Fig. S2A

Fig. S2. Silencing of HIF1 α in human islets suppresses PFKFB3 expression levels. (A) HIF1 α (upper panel) and PFKFB3 (lower panel) immunostaining of human islets transduced with LacZ-AdV (CTRL) or hIAPP-AdV for 48h with or without HIF1 α shRNA. HIF1 α or PFKFB3 is in red, insulin in green and nuclei in blue.

b. HIF1 α binds to PFKFB3 promoter and promotes PFKFB3 expression, yet increasing HIF1 α protein levels does not necessarily increase PFKFB3 protein expression. To show HIF1 α specificity and to establish a mechanistic connection as suggested in line 247, the authors need to do ChIP to show that HIF1 α binds to PFKFB3 HREs promoter in beta cell; they also need to silence HIF1 α in both rat (WT and HIP) and human islets (ND and T2D), perform qRT-PCR and WB to observe the changes in both PFKFB3 mRNA and protein levels.

Thank you for this suggestion. We used a luciferase reporter construct containing the PFKFB3 promoter with 2 hypoxia response elements (HREs) and tested its activation in cells transduced with LacZ or hIAPP expressing adenoviral vectors. Data are presented in Fig. 3A and show that luciferase activity is increased in cells transduced with hIAPP vector confirming the binding of HIF1 α to PFKFB3 promoter. HIF1 α silencing in non diabetic human islets transduced with hIAPP (figure above) reduces PFKFB3 expression in β -cells demonstrating PFKFB3 is under transcriptional control of HIF1 α .

Fig. 3. HIF1 α drives the expression of PFKFB3 in β -cells. (A) Luciferase assay showing the activation of PFKFB3 promoter containing 2 hypoxia elements. INS 832/13 overexpressing LacZ or hIAPP were transfected for 36h with plasmid vectors containing: RenSP luciferase gene without a promoter (empty vector-EV) measuring the background signal or housekeeping gene promoter driving the expression of RenSP luciferase gene (β -actin) or PFKFB3 promoter with hypoxia elements driving the expression of RenSP luciferase gene (PFKFB3). Data are presented as mean \pm SEM, n=9, *p<0.05.

5. The authors showed evidence hIAPP toxicity caused mitochondrial fragmentation and abnormal cytosolic calcium levels, and these phenotypes can be rescued through silencing PFKFB3. The authors suggest that silencing PFKFB3 should also improve insulin secretion, but they do not show these data. They need to perform GSIS in WT, hIAPP and hIAPP+PFKFB3 siRNA treated INS1 cells to make this conclusion.

We thank the reviewer for this helpful suggestion. PFKFB3 silencing did not rescue insulin secretion in INS 832/13 cells transduced with hIAPP but the INS cell models are not appropriate for insulin secretion studies. We therefore treated primary HIP rat islets with a published PFKFB3 inhibitor 3PO for 24h and this also does not restore the glucose induced insulin secretion assessed by perfusion (below). However, since this putative PFKFB3 inhibitor had a marked effect on insulin secretion in wild type islets, in which β -cells express minimal PFKFB3, we suspect that this inhibitor may have off target effects.

That said, given the metabolism data we would not expect PFKFB3 inhibition to restore glucose mediated insulin secretion. Specifically, the hIAPP toxicity induced decrease in β -cell pyruvate anaplerosis was not recovered by silencing

PFKFB3. The net outcome of this would be that the TCA intermediates originating from citrate derived through citrate synthase from oxaloacetate (via pyruvate carboxylase) and Ac-CoA (from pyruvate dehydrogenase) remain reduced, attenuating glucose stimulated insulin secretion as reported (Farfari S et al., *Diabetes*, 2000; Hasan NM et al., *J Biol Chem*, 2008; Jitrapakdee S et al., *Diabetologia*, 2010).

Fig. S9

Fig. S9. PFKFB3 inhibition does not restore insulin secretion in HIP islets. Measurements of glucose stimulated insulin secretion in vitro by perfusion system in islets from 5 months old WT and HIP rats treated or not with 3PO for 24 h. Dynamic insulin concentration during islet perfusion at low basal glucose (4 mM) (0-40 minutes) and high glucose (16 mM) (40-78 minutes) and high glucose with KCl (30 mM) (78-80 minutes). Data are presented as mean \pm SEM, n=2 for each group.

6. Quantitation for the Western Blots as shown in Fig. S2A (and throughout the manuscript) needs to be repeated. The authors should perform the assay for at least three independent trials. GAPDH is not an appropriate control for western blots where metabolism changes are also occurring as it is metabolically regulated.

Thank you. As requested the western blot now shows the protein levels of MFN-2, OPA-1 and Drp1 for three independent trials. Beside GAPDH, β -actin is used as loading control as suggested (Fig. S5A).

Fig. S4A

Fig. S4. hIAPP affects mitochondrial fusion by reducing MFN-2 levels but not mitochondrial fission. (A) Immunoblotting of indicated dynamin-related proteins in untreated (UT), control (CTRL, LacZ) and hIAPP overexpressing INS 832/13 cells.

7. Potential across the IMM is required for ongoing fusion. It is more likely that loss of potential in regions of the network (where MTR staining is lost) is the reason for the loss of fusion, not alterations in levels of MFN proteins.

Thank you for this suggestion. Flow cytometry profile of control and hIAPP transduced cells stained with TMRE show there is no difference in the mitochondrial membrane potential between these two groups (see above Fig. 5D). As a result, we conclude that the fragmentation of mitochondria caused by hIAPP is indeed a regulated adaptive change rather than a passive consequence of depolarised mitochondria.

Minor points

1. The authors adapted partial results from Schludi et al. in Fig. 3a, credit should be given in both text and figure legend (line 1071).

We have now cited Schludi in both the text and figure legend of Fig.1A.

2. The authors need to explain the doublet vs singlet bands seen in Cl. Casp-3 in Fig. 6C

The band of ~32 KDa corresponds to the full length caspase 3 whereas the band of ~22 KDa corresponds to the cleaved caspase 3. The Fig. 6C is not present anymore in the updated manuscript.

3. Figures S3e and 7a are repetitive; there is no need to present the same schematic figure twice.

Thank you for the comment; this has now been corrected.

4. The reason for the nomenclature change from hIAPP to HIP is unclear. The authors should also provide proper nomenclature of animal models used in their study.

Thank you for pointing this out. , We use hIAPP to refer to INS 832/13 cells transduced with an adenovirus expressing human IAPP, and HIP to refer to rats overexpressing human IAPP in their islet β -cells. hTG refers to mice overexpressing human IAPP in the β -cells of their pancreatic islets.

5. Change to number reference in line 329 – *This has now been changed.*

6. Typo in line 663, donor not donors – *This has now been changed.*

7. It should be Fig. 2a instead of Fig. 1a in line 1062 – *It has been changed.*

8. Unreadable characters in lines 493, 677, 679, 739 – *It has been corrected.*

Reviewer #2 (expert in IAPP)**(Remarks to the Author):**

The authors employ gene array and metabolomics approaches to identify potential regulators of human IAPP-induced toxicity in pancreatic beta cells. The findings link toxicity of human IAPP aggregates to mitochondrial disruption and glycolysis, in particular expression of PFKFB3. Of note, some of the histological findings are also found in pancreatic islets from type 2 diabetes. Although such findings are of some interest and novelty, the data as presented do not fully support the conclusions. A number of overstatements and broad conclusions are made, requiring 'connecting of the dots' such that their model diagram (Fig. 7) needs to be much better supported by data. The findings of changes in islet cell metabolism described in HIP rat islets may be secondary to hyperglycemia following loss of beta cell mass due to increased cell death from IAPP induced toxicity.

Thank you for the helpful and detailed review and constructive suggestions. As stated in the initial response to Reviewer #1, the helpful critiques not only led us into new experimental work, but resulted in an improved understanding of the data. We hope you will agree that the new appreciation that many of the apparently disparate manifestations of β -cell dysfunction in type 2 diabetes are explained by a stalled stress repair response that preserves cells at the expense of function is both novel and important. This also explains why the changes in neurons impacted by toxic oligomers so closely match those of β -cells in type 2 diabetes, and with both disease types progressing relatively slowly. As regards your comment about glucose induced changes, we clearly erred in not being sufficiently clear that the choice of age of the HIP rats was to precede diabetes and avoid changes consequent upon hyperglycemia. The blood glucose values in the rats we studied are shown in the table below (Supplementary Table 1), now added to the manuscript. Thank you for prompting this clarification.

Table 3. Characteristics of WT and HIP rats used for PFKFB3 protein expression study (Western blot)

ID	GENOTYPE	AGE (MONTHS)	BLOOD GLUCOSE LEVELS
366-5	WT	2	54
366-7	WT	2	58
367-15	HIP	2	67
367-3	HIP	2	61
357-3	WT	4	64
358-1	WT	4	65
357-7	HIP	4	78
359-3	HIP	4	70
346-5	WT	6	62
257-7	WT	6	77
443-4	WT	6	82
442-6	WT	6	79
346-7	HIP	6	73
259-3	HIP	6	84
442-1	HIP	6	90
443-2	HIP	6	72
443-3	HIP	6	79

Specific criticisms

1. Important conclusions are made based on changes in PFKFB3 expression that are not fully supported by the data provided:

(a) Timing of changes in PFKFB3 expression: the authors state that increased islet expression of PFKFB3 precedes onset of hyperglycemia at (6 months) in HIP rats. This could be important if true, but the data do not clearly show this. First, PFKFB3 mRNA levels at 6 months (when hyperglycemia is said to be present) are clearly elevated, but this is not obviously the case at 1 or 3 months (Fig 3D), where at each time point only one of two (only N=2) HIP bands appear denser. In a previous publication, blood glucose levels were elevated at 5 months, prior to any clear increase in PFKFB3 expression (based on the data provided). Importantly, blood glucose data from the current studies are not provided for comparison.

Thank you for this constructive criticism. We suspect that the publication you are referring to is that where HIP rats were fed a high fat diet that accelerated the onset of diabetes (Matveyenko AV et al., Diabetes, 2009) but fed a regular chow diet (as in the present study) in our facility developed diabetes at about 10 months of age (Matveyenko AV et al., Diabetes, 2006). Having provided the glucose values as requested for the rats in the present study it is apparent that "it is true" that the rats were not yet diabetic. To further emphasize the specific role of hIAPP toxicity in activating the HIF1 α -PFKFB3 pathway independently of glucose toxicity, it is notable that we first identified this activated pathway in the INS 823/13 hIAPP overexpressing cells that had no change in their exposure to glucose (Fig. S10A).

Fig. S10A

Fig. S10. hIAPP induced toxicity is linked to PFKFB3. (A) PFKFB3 protein levels in serum starved INS 832/13 after transduction with LacZ (CTRL) or hIAPP-AdV for 36h (upper panel) with or without PFKFB3 siRNA as assessed by western blot of whole cell extracts.

Secondly, the PFKFB3 protein data are described as being 2-fold greater *prior* to hyperglycemia but reported as 'not shown'. When there is supplementary data and therefore no reason to not show data, and in keeping with Nature Comm journal policy to avoid 'data not shown', and given that these data are important to the overall message of the importance of PFKFB3, these data should be provided and accurately described.

We apologize for this error. The data actually was shown in the prior manuscript, and is again shown in the revised manuscript in the Supplemental Fig. S1A-C.

Fig. S1A-C

Fig. S1. PFKFB3 is upregulated in prediabetic HIP rats of 3.5 months of age. (A) Representative Western blot of PFKFB3 in whole cell extracts from WT and HIP islets. **(B)** Representative immunofluorescence images of islets from WT and HIP rats at 2.5 (upper panel) and 3.5 (lower panel) months of age stained for PFKFB3 (red), insulin (green) and nuclei (blue). **(C)** Frequency of PFKFB3 positive β -cells in HIP vs. WT rats (2.5 and 3.5 months). Data are presented as mean \pm SEM, n=3 for each group, *p<0.05.

(b) Localization of PFKFB3 expression: By immunostaining, PFKFB3 immunoreactivity appears to be largely nuclear in HIP and T2D islets. The authors ascribe the changes in glycolysis in large part to the increase in PFKFB3 expression. Nuclear PFKFB3 has been described by others as a mechanism to increase cell proliferation in response to glucose.

Since glycolysis is presumed to occur in the cytoplasm, and how do the authors reconcile increased nuclear (but possibly not cytoplasmic) expression of PFKFB3 with increased glycolysis? In the model in Fig 7, the PFKFB3 is in the cytoplasm, not the nucleus, which seems an inaccurate depiction of the data. Is nuclear translocation of PFKFB3 occurring?

This critique raises several good points. First, yes, the reviewer is correct that the earlier version of the model diagram was in error and now has been replaced with a new schematic overview. Second, the reviewer is also correct that PFKFB3 is predominantly located in the nucleus. It is now well documented that metabolism is partially executed(?) in the nucleus (Boukouris AE et al., Trends Biochem Sci, 2016). In the cancer field, for instance, it is now well appreciated that regulation of cell cycle is mediated by not only the cyclins but also metabolites that act in the nucleus to signal that the cell is metabolically competent to execute cell replication. We have recently established that a similar pattern of metabolic changes is present during the cell cycle in β -cells (Montemurro C et al., Cell Cycle, 2017). The nuclear location of PFKFB3 has been ascribed to its recognised role to integrate the required increase in aerobic glycolysis that needs to precede successful cell replication with initiation of the cell cycle. PFKFB3 induced increased glycolysis is required to provide sufficient nucleotides for DNA synthesis as well as an alternative energy source to mitochondrial oxidative phosphorylation that is suspended while the mitochondrial network fragments in preparation for sorting to two daughter cells. PFKFB3 allosterically activates PFK1 via its product 2,6 biphosphate (F2,6BP), the activation of PFK1 in turn driving the increase in aerobic glycolysis. Meanwhile the metabolite F2,6BP also acts to signal initiation of cell replication. It is by this dual action of the product of PFKFB3 on metabolism (preparing a cell for cell cycle) and initiating cell replication, that PFKFB3 integrates preparing cells for cell cycle and then initiating that cell cycle. Finally, metabolites such as F2,6BP pass readily by diffusion through the nuclear pore complex, for example to activate cytosolic PFK1 without PFKFB3 requiring a cytoplasmic shuttle.

(c) Silencing of PFKFB3 expression: The authors show that silencing PFKFB3 restores mitochondrial fragmentation (Fig 6), beta cell survival and gene expression in INS-1 cells, but show no effects on glycolysis, cell respiration/OCR, lactate or ATP production or anything in primary islets. Such data are critical to the overall conclusions of the paper. It is also important to show from a therapeutic perspective that silencing PFKFB3 does not have deleterious effects (or perhaps improves) beta cell function (glucose-stimulated insulin secretion).

Thank you. As stated above we have substantially changed the “overall conclusions” of the paper away from the potential therapeutic benefit of inhibiting PFKFB3 to the insight that β -cell dysfunction in type 2 diabetes is a consequence of a stalled stress regeneration response. As such, we no longer advocate the inhibition of PFKFB3 as a therapeutic strategy. Indeed, while silencing of PFKFB3 does restore the mitochondrial network and decrease cell death, in studies prompted by the reviewers we now report that it does not restore insulin secretion in islets affected by toxic oligomers (please see Figure above). As regards metabolic changes in primary β -cells exposed to hIAPP toxicity, given the mixed population of cells in islets there is a practical barrier to undertaking metabolomics in primary β -cells, and in particular when the pathways concerned are potentially activated by anoxia of de-vascularized isolated islets. Given that the most prominent change in β -cell metabolism induced by the HIF1 α -PFKFB3 pathway was an increase in aerobic glycolysis most readily noted by increased lactate production, we have used lactate production to evaluate the metabolic actions of hIAPP toxicity and the suppression of the HIF1 α -PFKFB3 pathway in primary islets (see below). As well as being relatively readily measurable, the use of this approach has the added benefit that it has functional significance. In health, lactate generation is disallowed in β -cells since 100% of pyruvate must enter the TCA cycle to link the extracellular glucose concentration to ATP generation, closure of the K(ATP) channel and proportionate insulin secretion. Inappropriate β -cell lactate production is a characteristic of β -cells in type 2 diabetes (Zhao and Rutter, 1998; Martinez-Sanchez et al., 2015).

Fig. 6C

Fig. 6. PFKFB3 inhibition reduces lactate levels in HIP islets. (C) Lactate production rate measured in isolated islets from WT and HIP treated or not with 3PO for 24 h. Data are presented as mean \pm SEM, n=3 for each group, ***p<0.005.

2. The connection of HIF α and γ H2A.X to the observed changes are tenuous. γ H2A.X is said to be elevated in pre-diabetic HIP rats, although the data provided in Fig 3E are at 6 months when these rats are said to already be diabetic (though again, no glucose data are provided). It seems equally likely that the observed changes are secondary to human IAPP induced loss of β -cell mass (as is observed in HIP rats), and hyperglycemic stress on remaining β -cells. The authors state: "...these findings indicate that hIAPP decouples glycolysis from oxidative respiration due to γ H2A.X-associated accumulation of nuclear HIF1 α in INS 832/13 cells and β -cells from hIAPP transgenic rodents" These statements need to be softened or data provided; for example,

the temporal relationship of HIF1 α , PFKFB3, and γ H2A.x could be addressed in Fig 3D showing changes in HIF1 α and γ H2A.x protein over time.

Thanks, in this revised manuscript we have focused on the metabolism and mitochondrial network changes and removed the γ H2A.X data, focusing on the HIF1 α -PFKFB3 stress pathway inducing the metabolic changes. Having shown that the HIF1 α -PFKFB3 stress pathway is activated in hIAPP overexpressing INS823/13 cells with the anticipated downstream signals and metabolic changes we further established the specificity of this pathway to induce these metabolic actions by silencing the pathway in the setting of hIAPP toxicity. To extend these studies to primary β -cells, having established that hIAPP toxicity also induced the HIF1 α -PFKFB3 pathway in prediabetic hIAPP transgenic rats (Fig. 1A-E and Fig. S1), we silenced PFKFB3 in the setting of hIAPP toxicity and used the islet production of lactate as an indicator of aerobic glycolysis (see above). Since silencing PFKFB3 suppressed hIAPP induced islet lactate production in primary islets, it is reasonable to conclude that hIAPP induced disengagement of glycolysis from oxidative phosphorylation is mediated through induction of HIF1 α downstream target, PFKFB3. This conclusion is consistent with the known actions of PFKFB3 on metabolism (Najafov A and Alessi DR, PNAS, 2010).

3. While the authors conclude that hIAPP oligomers mediate the changes observed, this was not directly demonstrated, but rather appears to be based on their previous data showing oligomer formation in these models. They should either show oligomers histologically in cells associated with the metabolic changes and mitochondrial disruption, or more precisely state that the observations are associated with "human IAPP overexpression", or "human IAPP toxicity" or perhaps IAPP "misfolding" or "aggregates", since the species is not identified in these experiments. As examples change as in examples:

(a) Results section subtitle: "hIAPP toxic oligomers induce mitochondrial network fragmentation with reduced mitochondrial function."

(b) Discussion: "we uncovered that stress induced by hIAPP toxic oligomers recapitulates the metabolic phenotype reported in β -cells in T2D"

Changed to hIAPP induced toxicity as requested.

4. The array data and other studies are performed on whole islets, with conclusions that findings are occurring in β -cells. While the conclusion that these changes are occurring in β -cells is supported by complementary studies in INS-1 cells overexpressing human IAPP by adenovirus, it remains possible that changes in gene expression are occurring in non- β -cells. This should be clearly stated.

Good point, these limitations are added to results for islets.

5. Given the evidence (and conclusion) of a marked switch to aerobic glycolysis and production of lactate associated with human IAPP overexpression, and use of Seahorse to assess oxygen consumption, it would seem an easy, complementary, and useful measure for the authors to provide data of extracellular acidification rate (ECAR) from HIP and wild-type rat islets. Was acidification changed along with OCR and lactate production?

As requested, we have now also performed ECAR in hIAPP transgenic rat islets and the new data are now included Fig. S5A and B. These data confirm that ECAR was increased along with increased lactate production by HIP rat islets.

Fig. S5A-B

Fig. S5. Effect of hIAPP on extracellular acidification rate in rat islets. (A) Profiling of extracellular acidification rate (ECAR, mpH/min) in islets from WT and HIP rats measured with the Seahorse Bioscience XF24 extracellular flux analyzer. **(B)** Quantification of basal ECAR in islets of comparable surface area from WT and HIP rats. **** $p < 0.001$.

6. Fig 6D: human IAPP expression decreases cytosolic calcium and PFKFB3 increases it. To this reviewer, this does not make any sense in terms of the conclusions and the model in Fig. 7.

Thank you for pointing this out. Some of the confusion was due to an error in labelling, which has now been corrected. We hope the conclusions make more sense in the revision.

7. The TMRE data are also difficult to interpret as presented. First, there is no direct comparison between human IAPP and control cells, although it is said (data now shown) to be not different. Second, there is no shift in TMRE intensity, only a decrease in cell count. This suggests no difference in mitochondrial membrane potential (the driving force for ATP production) between human IAPP overexpressing and control beta cells, which seems surprising considering the conclusion that human IAPP overexpression is driving glycolysis, ATP production, and closure of potassium-sensitive ATP channels. One possible interpretation is that human IAPP overexpressing cells are more sensitive to cell death induced by any toxic stimulus (including DOG and oligomycin). Do the human IAPP expressing cells have increased sensitivity to any toxic stimulus? The TMRE data need to be interpreted in terms of their meaning to the proposed model of glycolysis and ATP production, since as presented the data do not seem to support this model. Ideally, ATP would be measured.

In response to this point, Figure 5D of the revision now includes the requested data (see above). The reviewer's conclusions are correct, and although we believe that when sustained, the mitochondrial membrane potential in hIAPP overexpressing INS 832/13 cells is vulnerable to secondary insults such as DOG and oligomycin (Fig. 5E, F).

Fig. 5E-F

Fig. 5. Mitochondrial respiration but not mitochondrial membrane potential is decreased by hIAPP. Mitochondrial membrane potential was measured by flow cytometry after labelling with TMRE dye in CTRL (**E**) or (**F**) hIAPP overexpressing cells in G1/S of the cell cycle in presence or absence of oligomycin (Oligomyc) or 2-deoxy-glucose (DOG).

Minor comments:

1. Fig S1 should clearly state what the control is, presumably rat IAPP adenovirus -

CTRL is AdV-LacZ and this is now explained in Fig. S3C-D.

2. Dotted lines should be used in Fig 7 for those aspects of the model that were not shown by the data in the current manuscript

The diagram has been changed with a new schematic presentation in Fig. 8B.

3. Why does PFKFB3 silencing increase G2/M – is it increasing cells in DNA damage checkpoint? Or increasing survival?

The increase in G2/M could be a result from a mitophagy checkpoint that is sensitive to PFKFB3 levels and aerobic glycolysis, and prevents mitosis in absence of PFKFB3 as previously reported (Domenech et al., Nat Cell Biol, 2015). Another possibility is that residual damage by hIAPP signals to the G2/M checkpoint despite PFKFB3 silencing which in replicating INS 832/13 cells diminishes most of the apoptotic signals driven by hIAPP.

**Reviewer #3 (expert in intracellular metabolism of beta cells)
(Remarks to the Author):**

The manuscript by Montemurro et al. assesses the mechanisms responsible for the toxic effects of human IAPP in β -cells. The authors perform studies mainly in insulinoma cells complemented with some studies in a rat model overexpressing hIAPP. Some studies using human islets are also included. The authors conclude that hIAPP induces major metabolic and mitochondrial network changes through the activation of the HIF1 α -PFKFB3 stress pathway. The authors propose that this pathway in the short term is aimed to preserve β -cell survival but in the long term result in cell death mediated by cytosolic Ca²⁺ accumulation.

1. The experimental designed is well thought out and the experiments are well designed. However, the findings in the study are for the most part confirmatory of data in the neuronal literature where similar processes occur in neurodegenerative disorders.

We appreciate that the reviewer's wide knowledge of the subject recognises the overlap between cellular dysfunction in protein misfolding diseases in the brain and in β -cells. Far from a being weakness, however, we respectfully submit that this overlap in what have been previously considered to be only weakly related diseases is one of the more important and exciting conclusions of the work we present in this manuscript. Furthermore, our manuscript far extends the molecular and metabolic consequences of protein misfolding by amyloidogenic proteins in the neurodegeneration literature. Taken together, the new findings have allowed us to put forward a completely novel model to account for the cellular dysfunction and gradual attrition of beta cells observed under these conditions. Specifically, we now propose that a highly conserved stress repair response pathway is activated by hIAPP, but in contrast to tissues having a regenerative capacity, both neurons and pancreatic β -cells, which have minimal replication capacity, as a result become trapped in a metabolic state that renders them dysfunctional while preserving them from immediate cell death.

2. In addition, many laboratories have shown that hIAPP induce β -cell dysfunction and death by inducing several mechanisms including oxidative stress, autophagy, mitochondrial dysfunction and ER stresses. How the current findings fit into the previous mechanisms is unclear.

Thank you for this question since our work has specifically contributed to that literature. Broadly speaking one can consider the literature on IAPP (and other amyloidogenic proteins such as Alzheimer's beta protein and synuclein) as falling into the following broad categories.

- (1) Why do these proteins aggregate in disease states? This can occur rarely because of mutations in the protein to increase amyloidogenicity or because mechanisms that prevent and defend against protein aggregation are overcome. We and others have noted the importance of the ubiquitin proteasome system, the autophagy pathway and fidelity of the ER unfolded protein response in defending against protein aggregation and toxicity.*
- (2) What is the aggregate form that is most toxic? We and others have demonstrated that the most toxic form IAPP oligomers appears to be small membrane permeant oligomers that most recent data suggest form within membranes.*
- (3) How is that toxicity manifest? This is the area with the widest variety of findings by different groups, perhaps not surprisingly since any loss of intracellular compartmentalisation due to unregulated membrane permeability would be predicted to induce manifest changes, including calpain hyperactivation and stress pathways.*

The major new insight that we present here addresses the following two questions,

- (1) Why is cell attrition so slow in diseases manifest by misfolded protein toxicity, such as β -cells in type 2 diabetes and neurons in Alzheimers disease? And,*
- (2) What is the basis for the major loss of function of these cells that precedes cell loss?*

We propose that the relatively long survival of the cells is due to activation of the HIF1 α stress/regeneration pathway, with the cells being trapped in the first stage of this pathway with metabolic adaptation that sustains injured cells. Further, we propose that the dysfunction of β -cells and neurons engaged (and trapped) in the "pro-survival stage 1" of this stress/regeneration pathway is a consequence of the functional dependence of these cell types on oxidative phosphorylation of glucose.

3. The authors use three models in these studies including INS cells, islets from HIP rats and human islets. The resilience in the use of INS1 cells without validation in the transgenic rat is concerning. There is no data on how the hIAPP is introduced to the cells and the cited reference does not include the description of the cells. This is important, as there are limitations of transformed cells INS1 (tumoral cells) that could potentially alter intracellular metabolism. In addition, it is

unclear if the degree of hIAPP overexpression between the INS1 model and the HIP islets is similar. No immunoblotting comparing levels of hIAPP was presented. Did the authors replicate findings by treatment with hIAPP?

We agree with the reviewer on the potential limitations of INS cell lines versus primary β -cells, as we have pointed out earlier in this Response to Critiques. We address the rationale for use of the various preparations we obtained data from in our point 3 to Reviewer #1. It has been our standard practice to require a consistent finding with these tools in order to feel comfortable to ascribe findings to being both due to hIAPP toxicity and potentially clinically relevant. In brief, the major findings presented in the present manuscript were first detected by unbiased RNA seq and metabolomics screening in the human IAPP expressing INS 823/13 cell model, corroborated in both human IAPP transgenic mice and rats and then human islets and human pancreas. We have previously provided the IAPP expression in each of the models used here, and to control for protein load in both the INS cell model and mice we have used soluble rodent IAPP as a control. We now include references specifically pointing out where methods in developing each of the models is stated, and where the relative IAPP expression levels for each is provided. We have not used extracellular addition of IAPP to induce toxicity for a long time now since the available data points to intracellular IAPP oligomers as being responsible for most toxicity in vivo although addition of protein that develops membrane permeant toxic oligomers extracellularly is not surprisingly also toxic (Janson J et al., Diabetes, 1999). The extracellular approach has been used to screen for compounds that inhibit aggregation and toxicity but these have invariably failed when scaled up to in vivo studies, emphasising the predominance of intracellular versus extracellular site of toxicity in vivo.

4. The authors focused on studying the effects of PFKFB3 as a mechanism for the phenotype. Interestingly, no quantification from the immunoblotting is presented and only two biological variables are included. Similarly, immunoblotting for PFKFB3 from human islets is not particularly convincing. Does hIAPP overexpression (as shown by the authors previously) in human islets reproduce this effect?? Given the numerous published mechanisms implicated in hIAPP induce β -cell dysfunction and apoptosis, it is unclear if this mechanism is a major component for regulation of β -cell mass in vivo.

Thank you, we have now added new western blots to quantify PFKFB3 expression in both humans and rodents (Fig. 1D see above) (Fig. 2D). With regards to the role of the HIF1 α -PFKFB3 pathway as an important mediator of β -cell loss in T2D, we agree that it is probably not primarily a mediator of cell death, but rather a more important mediator of β -cell dysfunction. Our more current understanding regarding the activation of this pathway is that it is protective at the expense of cell function. The best evidence to date for this, we believe, would suggest that calpain hyperactivation consequent to aberrant Ca²⁺ signalling is a more important mediator of amyloid protein induced cell death. While inhibition of PFKFB3 attenuates hIAPP mediated cell death in the INS cell model, the toxicity of hIAPP is accelerated as compared to that seen in primary β -cells. INS cells rapidly traverse the cell cycle and then undergo apoptosis at the G2/M checkpoint, whereas primary β -cells in adults much more rarely go through the cell cycle. This is the basis of our revised understanding that in non replicative cells, activation of the HIF1 α -PFKFB3 stress regeneration pathway results in cells becoming “trapped” in this pre-replicative first phase in which they are relatively protected from apoptosis at the expense of function.

Fig. 2D

Fig. 2. HIF1 α -PFKFB3 stress pathway is upregulated in β -cells from transgenic HIP rats and humans with type 2 diabetes (T2D). (D) Representative Western blot of PFKFB3 and HIF1 α levels in nuclear enriched- and whole cell extracts from non-diabetic (ND) and T2D donor islets.

5. Mitochondrial function can be regulated by many mechanisms. The authors show mitochondrial dysfunction at 4-6 months of age (4 or 6m?). Do this occur earlier?

Thank you for giving us the opportunity to clarify a misunderstanding that has been brought up in several of the reviewers' questions and indicates we did not clearly enough explain the HIP rat model. When this animal model of type 2 diabetes, is fed a regular chow diet, as in the presented experiments, the rats are prediabetic at 4 months of age but have a β -cell defect and a measurable increase in β -cell apoptosis (Matveyenko AV, Diabetes, 2006). We chose this age to study in order to identify the early consequences of protein misfolding before the onset of frank diabetes, which occurs at around 10 months of age when this model is fed normal chow. The rate of onset of diabetes in the HIP rat model is accelerated by a high fat diet (Matveyenko AV, Diabetes, 2009), but we did not use a high fat diet in our study.

Why does it take 4-5 months to develop if hIAPP is expressed from development? Therefore, the authors should perform mitochondrial function studies at earlier stages to provide stronger evidence for this mechanism in the phenotype.

Thank you for the question. Our understanding of altered mitochondria function in response to hIAPP toxicity has changed considerably since the first version of this manuscript was submitted as we now more fully appreciate that the altered mitochondrial network and disengagement of glycolysis from the TCA cycle we reported originally are manifestations of activation of the HIF1 α -PFKFB3 stress/regeneration pathway. The fact that mitochondria in cells with hIAPP toxicity retain their membrane potential and that this same mitochondrial morphological adaptation (network fragmentation and perinuclear location) has recently been reported in response to Alzheimer beta toxicity in neurons as a defence against Ca²⁺ toxicity, further implies that the hIAPP induced changes are pro-survival adaptive rather than mediators of toxicity. Formation of toxic protein oligomers from amyloidogenic proteins occurs when the rate of protein synthesis overcomes the cells capacity to fold the majority of newly synthesized proteins and remove any misfolded proteins. This threshold can be approached in healthy cells by sufficiently increasing expression (the hIAPP INS 823/13 cell model). While there are rare examples of mutant amyloidogenic proteins, the mutation increasing amyloidogenicity and leading to young onset disease (example hereditary Dutch Alzheimer's disease), the majority protein misfolding diseases including T2D, Alzheimer's, Parkinson's are diseases of aging. At birth and during childhood there is a huge capacity for protein synthesis and folding, as required by the massive protein synthetic burden required for growth of the individual. Chaperone proteins, the proteasome and autophagy are all highly upregulated at birth and through childhood. Once adulthood is reached there is a progressive decline in chaperone protein availability and available proteosomal and autophagy flux. As a result, non replicative cells that express amyloidogenic proteins with a high protein synthetic burden (neurons and β -cells) become progressively more vulnerable to protein misfolding. This plays out in our various hIAPP transgenic rodent models. Hemizygous hIAPP mice do not develop diabetes unless we induce increased hIAPP expression through insulin resistance. Alternatively, we can increase hIAPP expression by cross breeding to homozygosity and this leads to spontaneous diabetes. In the HIP rat model we can accelerate diabetes with high fat feeding inducing insulin resistance.

There are mechanisms to prevent toxicity resulting from the expression of amyloidogenic proteins (chaperone proteins, the ubiquitin proteasome system, the autophagy lysosomal pathway) and presumably this is why the majority of humans (all of who express human IAPP in β -cells) do not develop diabetes. Even in those humans that do develop type 2 diabetes, in common with neurodegenerative diseases, the incidence of type 2 diabetes increases progressively with aging. The function of pathways which are protective against protein misfolding also decline with age. In addition, insulin sensitivity declines with age so that the expression of IAPP increases with age. Taken together, the formation of IAPP toxic oligomers and associated toxicity will occur when the cellular capacity to prevent oligomer formation is overcome by IAPP expression.

The authors study fusion/fission in INS1 cells and function in islets.

Thank you. In our experience it is hard to obtain sufficient numbers of sorted β -cells from isolated islets (particularly from islets with amyloid present, as in the HIP rat) where the cells are functional enough to perform studies of mitochondrial fission and fusion. Nearly all of the β -cell mitochondrial network turnover work has thus been done using INS1 cells rather than islets. Moreover, we had a clear documented impact of hIAPP toxicity on the kinetics of mitochondrial network changes in the INS cells that we could interrogate on the mechanism of network fragmentation. On the other hand, we were able to perform Seahorse studies in primary islets which are more glucose responsive than INS1 cells and therefore more appropriate to evaluate the mitochondrial response to an increase in glucose.

6. The metabolic studies are exclusively performed in INS1 cells. Are these defects reproduced in islets beyond lactate levels. Key metabolic flux findings should be replicated in islets from HIP model. Similarly, calcium measurements should also be performed in islets from HIP rats.

Thank you. The problem with isolated islets and the metabolic pathways are several. First, since the percentage of the islet cells that are β -cells is less in HIP rat islets than WT islets and so the metabolic study findings may reflect altered cellular composition rather than β -cell metabolism. Second, the inner core of isolated islets is relatively anoxic after isolation and removal from blood supply, and we are concerned as to how this would impact interpretation of metabolism. Therefore, for these reasons we chose to focus on the specific impact of hIAPP toxicity in the INS1 cell model where we could isolate the changes in metabolism from these uncontrolled variables. We selected lactate production in response to glucose as the most well characterised alteration in β -cells in T2D that was then consistent with findings in the HIP rat and INS1 cells. As requested we perform calcium measurements in islets and included in whole islets as shown in Fig. 7C.

Minor

1. Reference for INS1 cells overexpressing hIAPP is wrong –*this reference has now been corrected.*
2. Table with more information on islet donors is missing. Diabetes duration, etc?? –*The tables below are now included.*

Table 4. Characteristics of islet non diabetic donors used for HIF1 α and PFKFB3 protein expression study (Western blot)

ID	SEX	AGE (years)	BMI	CAUSE OF DEATH
HI126	M	47	31	Anoxia
HI131	F	24	32	Anoxia
HI129	M	23	25	Anoxia

Table 5. Characteristics of islet diabetic donors used for HIF1 α and PFKFB3 protein expression study (Western blot)

ID	SEX	AGE	BMI	DURATION OF DIABETES (years)	TREATMENT	CAUSE OF DEATH
HI128	M	57	34	8	Untreated	Head trauma
HI130	M	47	32	>10	Diet, oral medications	Cerebrovascular/stroke
HI132	F	56	25	0-5	Diet, oral medications	Cerebrovascular/stroke

Reviewer #2 (Remarks to the Author):

The authors have addressed this reviewers' comments and the manuscript is much improved, particularly concerns about hyperglycemia driving the observed changes in gene expression. Considerable new data are provided, as well as some changes to the interpretation, which is appropriate.

The phenotype of the HIP beta cells resemble in some ways those of dedifferentiated beta cells and it would be worthwhile, given recent interest in beta cell dedifferentiation in T2D, that the authors briefly mention in the discussion whether their cells are indeed demonstrating a dedifferentiated phenotype or something distinct.

Reviewer #3 (Remarks to the Author):

All my concerns have been addressed.

Reviewer #4 (Remarks to the Author):

Authors have performed additional experiments to answer the questions raised by reviewer, which significantly improves manuscript. Their findings about increased HIF-1 α mediated stress response in hIAPP model as well as T2 diabetes is clearly unbiased and robust. Authors suggest that, by activating HIF-1 α -PFKFB3 pathway in response to hIAPP, beta cells lost their major functions including glucose-stimulated insulin secretion as the price of life preservation. However, still major concern remains whether upregulation of HIF-1 α -PFKFB3 pathway acts a critical role in protecting against IAPP-mediated toxicity rather than just one of parallel stress responses. The provided evidence in this study showing a pro-survival role PFKFB3 is not sufficient to support the main conclusion. Furthermore, the protective role of mitochondrial fragmentation by PFKFB3 upregulation should be demonstrated. As the major mechanism for fragmentation suggested by authors, suppression of mitofusin-2 expression has serious detrimental effects on mitochondrial functions reported by many publications. Their Ca²⁺ concentration measurements (cyto, mito, ER) comparing the basal ratio of fluorescence (Fura2, RP, D4ER) is not also convincing. I think these points critically weaken the importance of main findings of this study and make to us overemphasize the significance and conclusion.

RESPONSES TO REVIEWERS' COMMENTS

Reviewer #2 (Remarks to the Author):

The authors have addressed this reviewers' comments and the manuscript is much improved, particularly concerns about hyperglycemia driving the observed changes in gene expression. Considerable new data are provided, as well as some changes to the interpretation, which is appropriate.

The phenotype of the HIP beta cells resembles in some ways those of dedifferentiated beta cells and it would be worthwhile, given recent interest in beta cell dedifferentiation in T2D, that the authors briefly mention in the discussion whether their cells are indeed demonstrating a dedifferentiated phenotype or something distinct.

Thank you for helpful prior review that indeed guided us to much greater insights into the adaptive state of β -cells in response to hIAPP toxicity, and in particular to separate glucotoxicity from hIAPP toxicity. Your suggestion to link the findings to the dedifferentiation concept is an excellent one, indeed the present data are actually a first mechanism for the widely reported descriptive dedifferentiation of β -cells, again thank you. We had a single sentence on this in the prior manuscript, but have elaborated on this as suggested as below.

While β -cells in adults have a limited capacity to complete cell cycle, the partial dedifferentiation of β -cells previously reported in T2D^{69,70,71} may reflect the sustained signaling for entry into, but failure to execute, cell cycle. Partial dedifferentiation is a regulated step in preparation for replication by differentiated cells such β -cells⁷². The adaptive changes in metabolism and mitochondrial network induced by the HIF1 α /PFKFB3 pathway in response to hIAPP toxicity are also present in β -cells in T2D and are comparable to those present in replicating β -cells⁷³. Immature β -cells retain comparable metabolism presumably to permit cell replication, so it is not surprising that β -cells exposed to the sustained HIF1 α /PFKFB3 pathway might be considered as adopting an immature dedifferentiated status. The current study provides a plausible mechanism for that process and establishes that it is protective of β -cell viability against stress at the expense of β -cell function.

Reviewer #3 (Remarks to the Author):

All my concerns have been addressed. *Thank you also for your helpful and constructive critiques that along with the other reviewers were most constructive in driving our subsequent experiments and insights.*

Reviewer #4 (expert in beta cell metabolism)

Authors have performed additional experiments to answer the questions raised by reviewer, which significantly improves manuscript. Their findings about increased HIF-1 α mediated stress response in hIAPP model as well as T2 diabetes is clearly unbiased and robust. Authors

suggest that, by activating HIF-1 α -PFKFB3 pathway in response to hIAPP, beta cells lost their major functions including glucose-stimulated insulin secretion as the price of life preservation.

1. However, still major concern remains whether upregulation of HIF-1 α -PFKFB3 pathway acts a critical role in protecting against IAPP-mediated toxicity rather than just one of parallel stress responses. The provided evidence in this study showing a pro-survival role PFKFB3 is not sufficient to support the main conclusion. Furthermore, the protective role of mitochondrial fragmentation by PFKFB3 upregulation should be demonstrated.

Thank you for this excellent suggestion. Fortunately there is a highly specific and well characterised HIF1 α inhibitor (<http://www.selleckchem.com/products/kc7f2.html>) and an efficient PFKFB3 siRNA to address this question.

Having confirmed the efficacy of the HIF1 α inhibitor in pancreatic β -cells, we investigated the impact of inhibition of HIF1 α in beta cells expressing human IAPP. As shown in figure 8 and supplementary figure S10, inhibition of HIF1 α in the setting of hIAPP expression resulted in increased cell death as documented by both increased caspase-3 on western blotting, increased TUNEL by immunostaining and an increased frequency of sub-G1 cells by FACS analysis indicating the DNA fragmentation characteristic of apoptosis. Furthermore, we then also investigated the impact of inhibition of PFKFB3 in β -cells expressing human IAPP. Again, as shown in figure 8 and supplementary figure S10, these studies confirmed the beta cell protective role of PFKFB3 under conditions of hIAPP toxicity since inhibition of PFKFB3 also resulted in increased beta cell death determined by caspase-3, TUNEL and FACS analysis.

Therefore, we have established that the activation of the HIF1 α /PFKFB3 pathway by hIAPP toxicity is indeed protective against beta cell death as suggested. Again thank you for this excellent suggestion.

2. As the major mechanism for fragmentation suggested by authors, suppression of mitofusin-2 expression has serious detrimental effects on mitochondrial functions reported by many publications.

We fully understand this question of the reviewer as our own starting point was to assume that the altered mitochondrial network we noted in response to hIAPP was a mediator of toxicity. This assumption by us came from the publications the reviewer refers to in which acute fragmentation of mitochondria is shown to mediate the mitochondria pathway of apoptosis. In that form of mitochondrial fragmentation there is loss of integrity of the mitochondrial membranes and release of cytochrome C.

However, we have come to appreciate that there is a longer term adaptive form of mitochondrial network change towards a fragmented (but membranes intact and no cytochrome C leakage) perinuclear form (versus dispersed reticular form) that occurs both in preparation for cell cycle (Yamano K, Coupling mitochondrial and cell division. Nat Cell Biol. 2011, 13:1026-1027) and in long lived cells such as neurons in Alzheimer's disease and beta cells in type 2 diabetes as a

component of a defense response (Moreira PI et al. The key role of mitochondria in Alzheimer's disease. *J Alzheimers Dis.* 2006, 9:101-110). Of interest, not only is the form of the mitochondrial network comparable in cancer cells and both neurons and beta cells subject to misfolded protein stress, but glucose metabolism is also comparable under these two conditions, with high glycolytic flux with pyruvate diverted to lactate rather than the TCA cycle, with high pentose phosphate pathway flux for DNA synthesis (cancer) or repair (pro-survival in neurons and beta cells), and mitochondria engaged in synthesis of biomass (for new daughter cells or repair) rather than oxidative phosphorylation.

This adaptive perinuclear form of mitochondria network has recently been shown to protect against the toxic effect of aberrant cytosolic Ca surges characteristic of misfolded protein stress in neurons by attenuating mitochondrial Ca uptake (as noted in beta cells in the present manuscript) and the subsequent induction of mitochondrial induced apoptosis. (Szabadkai G et al. Drp-1-dependent division of the mitochondrial network blocks intraorganellar Ca^{2+} waves and protects against Ca^{2+} -mediated apoptosis, *Mol Cell.* 2004, 16, 59-68). Adaptive mitochondrial fragmentation by the mechanism of attenuated mitofusin-2 has also been reported as a regulated mechanisms subserving the activation of T-cells to a pro-replicative state (Dasgupta A. Mechanism of Activation-Induced Downregulation of Mitofusin 2 in Human Peripheral Blood T Cells. *J Immunol.* 2015, 195(12):5780-6) and in regulated adaptation of brown fat cells to withstand the stress of insulin resistance (Mahdaviani K et al. Mfn2 deletion in brown adipose tissue protects from insulin resistance and impairs thermogenesis. *EMBO Rep.* 2017 18(7):1123-1138). Interestingly in each of these circumstances the pattern of metabolic adaption mirrors that which we have found with hIAPP induced beta cell toxicity, and has previously been reported in beta cells in humans with type 2 diabetes and impacted neurons in Alzheimer's disease.

In the comments to the editor we note that there seems to be a little confusion as to the purpose of the DRP-1 dominant negative study. To clarify, this was simply done to confirm that hIAPP toxicity does not increase mitochondrial fragmentation by enhanced DRP-1 mediated fission but rather only through decreased MFN2 induced fusion. The answer was clear, not only were DRP-1 levels not increased but the adaptive changes in mitochondrial network form occurred even in the presence of the DRP-1 dominant negative inhibition, so the changes were clearly MFN2 mediated. This is then consistent with the mechanisms subserving adaptive fragmentation of the mitochondrial network cited above. To that point in comments to the editor you also state, "It is questionable whether Mfn2 upregulation can rescue IAPP-mediated aberrant Ca^{2+} changes and beta cell toxicity". We agree, and did not intend to imply that this is the case. To the contrary as hopefully we have now clarified, we view the adaptive changes in mitochondrial network mediated by MFN2 as protective.

Taken together, we agree completely with the reviewer that acute induction of mitochondrial fragmentation with MFN-2 depletion can be pro-apoptotic but we have come to appreciate that it can also be a long term adaptive (cancer, activated T-cells, stressed adipose cells, stressed beta cells and stressed neurons). We accept that we did not make this distinction sufficiently in the prior discussion and now add the following paragraph.

Mitochondria network form and disposition is regulated by alterations in the balance of network fusion and fission. The more fragmented form in response to hIAPP was mediated by a decrease in the fusion protein MFN2, mirroring the mechanism subserving adaptation to a more fragmented network form in response to stress in neurons³² and activation of T-cells⁷⁴ and in regulated adaptation of brown fat cells to withstand the stress of insulin resistance⁷⁵. The more fragmented perinuclear mitochondrial network has been shown to be protective against the potentially deleterious effects of aberrantly high cytosolic Ca²⁺ waves^{32, 76} as present in hIAPP toxicity, and following ischemic reperfusion injury in cardiomyocytes⁴⁸. Of interest the latter are also protected from cell death by activation of the HIF1 α stress pathway⁴⁸.

3. Their Ca²⁺ concentration measurements (cyto, mito, ER) comparing the basal ratio of fluorescence (Fura2, RP, D4ER) is not also convincing. I think these points critically weaken the importance of main findings of this study and make to us overemphasize the significance and conclusion.

And in comments to editors... 'In addition, they suggest that there is an increased cytosolic Ca²⁺ and a reduction of ER-mito Ca²⁺ transfer as consequences of HIF1 α -PFKFB3 activation. I would recommend to show not only basal level but also glucose-stimulated cytosolic/mitochondrial calcium changes. For ER calcium level, I am not sure they tried SERCA inhibitor and got the differences from the resting level, which reflect ER Calcium content.'

We apologize if it was not sufficiently clear, these measurements were all performed under stimulatory conditions (glucose 11mM) as previously stated in methods but now also in the results section.

Regarding the technical aspects of the Ca²⁺ measurements, we chose to directly monitor ER free Ca levels using D4ER, a genetically encoded and ER targeted FRET probe. Expression was beta cell specific via the use of RIP2 promoter. This probe has been previously verified in pancreatic beta cells by Patrick Gilon's group in Brussels, who confirmed that it is not only properly targeted to the ER but reports robust changes in ER Ca under a wide variety of experimental conditions (i.e. responses to ACh stimulation, ER depletion due to thapsigargin, etc; see Ravier MA et al. Mechanisms of control of the free Ca²⁺ concentration in the endoplasmic reticulum of mouse pancreatic β -cells: interplay with cell metabolism and [Ca²⁺]_c and role of SERCA2b and SERCA3. Diabetes. 2011, 60(10), 2533-2545.

Measuring ER free Ca²⁺ directly instead of indirectly by cytosolic changes to thapsigargin is superior for 3 reasons: 1) Cytosolic Ca²⁺ responses to ER depletion depend on the presence and size of store operated Ca current (SOC) and cannot distinguish between Ca²⁺ rises from influx vs. Ca²⁺ release. 2) Blocking SERCA Ca²⁺ATPases in the ER will reduce ER Ca²⁺ and cause a transient increase in cytosolic Ca²⁺ as a result, but will not affect steady state cytosolic Ca²⁺ due to the plasma membrane Ca²⁺ATPase. 3) In our hands, Thapsigargin-induced rises in cytosolic Ca²⁺ are produced but do not reliably reflect the size of the ER Ca²⁺ store, either for the reasons stated above and/or a lack of sensitivity. For these reasons we are confident that monitoring ER Ca²⁺ directly is the best approach overall.

Reviewer #2 (Remarks to the Author):

The authors have satisfactorily addressed my concerns. The manuscript will make a nice contribution to the field

Reviewer #4 (Remarks to the Author):

The authors successfully performed additional experiments answered the comments raised by reviewer. However, question about Ca²⁺ measurement was misunderstood and answered incorrectly. They did not answer to the points written in comments to editors (I would recommend to show not only basal level but also glucose-stimulated cytosolic/mitochondrial calcium changes. For ER calcium level, I am not sure they tried SERCA inhibitor and got the differences from the resting level, which reflect ER Calcium content.'). Particularly, ER Ca²⁺ measurement, it is required to show the difference in ratio between the basal level and the Thapsigargin (or CPA)-induced depleted level.

Point-by-point responses to the reviewers' critiques

Reviewers 1, 2 and 3.

We thank the editors and reviewers for their constructive guidance. We are grateful to reviewers 1, 2 and 3's support for publication after the additional experiments that they proposed strengthened the manuscript. We have further revised the manuscript and addressed the additional suggestions by reviewers as below

Reviewer #2 (Remarks to the Author):

The authors have satisfactorily addressed my concerns. The manuscript will make a nice contribution to the field

Response. Thank you.

Reviewer 4. (Remarks to the Author):

(1). "The authors successfully performed additional experiments answered the comments raised by reviewer.

Response. We thank the reviewer for acknowledging that we indeed successfully performed the additional new studies requested by the reviewer to demonstrate that the inhibition of the HIF1 α /PFKFB3 signaling pathway increased beta-cell vulnerability to human IAPP induced cytotoxicity. We agree that the added experiments the reviewer suggested strengthened the manuscript.

(2). However, question about Ca²⁺ measurement was misunderstood and answered incorrectly. They did not answer to the points written in comments to editors (I would recommend to show not only basal level but also glucose-stimulated cytosolic/mitochondrial calcium changes.

Response. We apologize for our apparent oversight in not addressing these points. As requested, we now show both the basal (2.8 mM glucose) and glucose stimulated (16.8 mM glucose) Ca²⁺ data for each compartment and under each of the canonical experimental conditions, as requested, in the new Figure 8. In addition, we also show representative Ca²⁺ traces of individual INS 832/13 cells as further documentation, as part of our new Supplemental Figure 9. We have also amended the text of the Results section and the figure legends accordingly, with regards to the new Ca²⁺ results. We thank the reviewer for clarifying his/her request and for making these important points.

(3). For ER calcium level, I am not sure they tried SERCA inhibitor and got the differences from the resting level, which reflect ER Calcium content). Particularly, ER Ca²⁺ measurement, it is required to show the difference in ratio between the basal level and the thapsigargin (or CPA)-induced depleted level.

Response. We regret that we apparently misunderstood the reviewer's suggestions at first regarding the Ca²⁺ measurements.

In response to Reviewer 4's critique, we carried out new experiments to probe more deeply into the actions of hIAPP toxicity and the adaptive pathway on the Ca^{2+} pools under study here, rather than simply monitoring steady-state levels of Ca^{2+} as we did before.

In our new Fig. 8 and supplementary Fig. S9, we now show how the mean Ca^{2+} levels of the cytosolic, ER, and mitochondria pools change over time after changing extracellular glucose from 2.8 to 16.8 mM glucose in INS832/13 cells exposed to hIAPP toxicity and amelioration by PFKFB3 inhibition, which provides a significant expansion of our data set and more insight into the impact of hIAPP toxicity and the consequent expression of PFKFB3 on calcium handling by the beta-cells.

While INS cells maintained low cytosolic Ca^{2+} levels in 2.8 mM glucose, cytosolic, ER, and mitochondrial Ca^{2+} levels increased in response to raising glucose to 16.8 mM under control conditions, as expected. Under conditions of hIAPP toxicity, INS1 cells exhibited elevated basal cytosolic Ca^{2+} but still responded to a rise in glucose. However, with hIAPP toxicity there was a profound effect on both ER and mitochondrial Ca^{2+} levels, increasing the basal levels to what appeared to be saturated levels and severely blunting subsequent responses to elevated glucose. Despite being unaffected by raising glucose with hIAPP toxicity, the ER and mitochondrial Ca^{2+} levels were subsequently decreased by application of the SERCA-inhibitor cyclopiazonic acid (CPA), or the mitochondrial inhibitor sodium azide, respectively. This confirms that these organelles were still capable of sequestering Ca^{2+} in an energy-dependent manner.

Importantly, repeating the measurements again with comparable hIAPP expression but concurrent silencing of PFKFB3, the basal Ca^{2+} levels of all three compartments was restored as was the increment in Ca^{2+} to an increase in glucose concentration, strongly supporting the main hypothesis of this study.

The current data support a model whereby hIAPP toxicity facilitates Ca^{2+} entry into the cell, most likely by facilitating closure of K(ATP) channels even at basal glucose concentrations due to the sustained high flux through glycolysis (and thus ATP generation) consequent upon the $\text{HIF1}\alpha/\text{PFKFB3}$ induced pro-survival remodeling of metabolism. This can then explain the elevated basal cytosolic Ca^{2+} while the saturation of the ER and mitochondrial pools likely reflect the effects of hIAPP toxicity to increase SERCA activity, to accommodate the sustained influx of Ca^{2+} into the cytosol.

We have now further clarified the Experimental Methods as well as Results sections of the paper to more clearly and completely present the new data found, and to provide more details as to the methods used and their limitations. We thank Reviewer 4 for the valuable insights and suggestions which permitted us to significantly improve the paper in these ways.

Reviewer #4 (Remarks to the Author):

Satisfied with new data in Fig. 8 and Fig. S9